# Insights into the structure and initial host attachment of the flagellotropic bacteriophage 7-7-1

W. E. M. Noteborn [1,7], R. Ouyang [2,3,7], T. Hoeksma [3,7], A. Sidi Mabrouk [3], N. C. Esteves[4],
D. M. Pelt [5], B. E. Scharf [4] & A. Briegel [3,6] ✉

Understanding the structural and functional mechanisms of bacteriophage 7-7-1, the flagellotropic phage infecting *Agrobacterium* sp. H13-3, offers promising insights into phage-host interactions. Using single particle analysis (SPA) cryo-electron microscopy (cryo-EM), we determined the capsid, neck region, tail, and baseplate complex structures. Combined with cryo-electron tomography (cryo-ET) and machine learning methodologies, our findings indicate that phage 7-7-1 uses capsid fibers to establish initial contact with the host flagellum, followed by subsequent attachment to cell surface receptors. The study also demonstrated that capsid fibers are flexible and can interact with other phages and host flagella, suggesting a cooperative infection strategy. These results provide crucial structural insights and may open avenues for developing phage-based therapeutics against resistant bacterial pathogens.

Flagellotropic bacteriophages represent a distinct group of phages that initiate their infection cycle by attaching to the flagellum of their motile host. The bacterial flagellum, a helical extracellular appendage powered by a rotary motor used for swimming and swarming, serves as the initial point of attachment for these phages. Exploiting the rotational motion of the flagellum, flagellotropic phages navigate towards the bacterial cell surface, where they establish interactions with secondary receptors and ultimately deliver their genetic material into the host cytoplasm. The infection process of flagellotropic phages involves intricate and diverse structural features that enable them to withstand substantial drag forces and torques. In-depth studies have shed light on the molecular mechanisms and dynamic aspects of flagellotropic phages and their interactions with host organisms. Some examples of previously studied phages include the phage F341[1] that infects *Campylobacter jejuni*, and the phage PBS1[2] that infects *Bacillus* species. Another example is the phage χ[3], which infects various genera of *Enterobacterales*, presumably by utilizing its tail fiber to bind to the flagellum[4]. Other phages, such as ΦCbK[5], use long capsid fibers to attach to the flagellum and also require active rotation to reach the surface of the host.

*Agrobacterium*, a phytopathogenic bacterium, utilizes horizontal gene transfer to induce crown gall disease in plants[6]. Activation of its virulence gene complex, in response to plant signals, facilitates the transfer of the

T-DNA segment from its Ti-plasmid into the plant genome[7]. Notably, *Agrobacterium* can also infect humans and other animals, particularly those with compromised immune systems, leading to opportunistic infections such as *sepsis*, *monoarticular arthritis*, *bacteraemia*, and *endocarditis*[8]. Moreover, it can produce harmful hydrogen sulfide gas, causing respiratory irritation and damage[9]. *Agrobacterium* has also been linked to certain human diseases, such as cancer and Morgellons disease[10].

Here, we studied the unusual architecture of the lytic flagellotropic phage 7-7-1 and its interaction with its non-infectious host *Agrobacterium* sp. H13-3[11]. Phage 7-7-1 belongs to the *Myoviridae* family[12] and was isolated from compost soil in Germany[13]. This phage contains double-stranded DNA of 247,374 base pairs[14,15], exhibits high sequence similarity to *Agrobacterium* phage Milano[16] and *Agrobacterium* phage OLIVR4[17] (80% similarity to both determined by ANI), including an unusual accumulation of 1.45% cysteine residues in the total proteome. In comparison, the T4 and A511 phages have a cysteine content of 0.9% and 0.7%, respectively. Typical for *Myoviridae*, it also contains a long contractile tail with multiple elongated and kinked tail fibers[13,18,19]. Phage 7-7-1 uses the host lipopolysaccharide as a secondary cell surface receptor[20], and its life cycle encompasses an eclipse period of approximately 60 min, with complete phage propagation achieved within 80 min. The estimated burst size

[1]Netherlands Centre for Electron Nanoscopy, Leiden University, Leiden, The Netherlands. [2]MOE Key Laboratory for Nonequilibrium Synthesis and Modulation of Condensed Matter, School of Physics, Xi'an Jiaotong University, Xi'an, China. [3]Institute of Biology, Leiden University, Leiden, The Netherlands. [4]Virginia Polytechnic Institute and State University, Blacksburg, VA, USA. [5]Leiden Institute of Advanced Computer Science, Leiden University, Leiden, The Netherlands. [6]Integrative Structural Cell Biology Unit, Department of Structural Biology and Chemistry, CNRS UMR 3528, Institut Pasteur, Paris, France. [7]These authors contributed equally: W. E. M. Noteborn, R. Ouyang, T. Hoeksma. ✉e-mail: ariane.briegel@pasteur.fr

of phage 7-7-1 amounts to 120 particles per bacterial cell[18]. The most unusual characteristic of this phage is the multiple capsid fibers that appear to emerge from the icosahedral capsid. While capsid fibers have been studied in other viruses and phages previously[21–23], they remain highly challenging subjects for structural studies due to their often-flexible nature. The unusual appearance and putative function of the capsid fibers of phage 7-7-1 are highly intriguing, but so far, only low-resolution negative stain electron micrograph images are available[18].

To gain insight into the unique structural characteristics of phage 7-7-1, we employed different structural methodologies to investigate its structure. Through single particle analysis (SPA) cryo-electron microscopy (cryo-EM), we generated atomic models of the well-organized phage capsid, neck region, tail, and baseplate complex. In addition, we determined the presence of capsid fibers, with one individual fiber emerging from each vertex of the capsid. To gain further insight into the structure of these flexible capsid fibers, we employed cryo-electron tomography (cryo-ET). Leveraging machine learning techniques, we successfully used a trained neural network to automatically track the intricate capsid fibers, facilitating quantitative analysis of the tomography data. Our findings highlight the robust attachment capability of the 7-7-1 capsid fibers to flagella during the infection process. Through the synergistic integration of structural, computational, and experimental approaches, we have achieved an in-depth understanding of the structure and functionality of this distinct bacteriophage.

## Results
### Capsid structure of phage 7-7-1
To gain insight into the structural characteristics of the phage 7-7-1 capsid, we chose to use cryo-electron microscopy (cryo-EM) (data collection

parameters and model building statistics in Tables 1 and 2, respectively). Cryo-EM micrographs of phage 7-7-1 reveal the typical structure of this phage: the large icosahedral capsid is coupled to a contractile 135 nm long tail that is intricately adorned with an abundance of bushy tail fibers. A defining characteristic of phage 7-7-1 is the multiple long and curly capsid fibers surrounding each capsid (Supplementary Fig. 1, white arrows). Our dataset reveals two distinct variants of the capsid: full capsids containing DNA and empty capsids (Supplementary Fig. 1). Notably, most of the phages observed in the micrographs are intact capsids filled with DNA. To gain detailed insights into the capsid structure, we applied single particle analysis (SPA) and obtained a 3.39 Å structure of the intact, DNA-containing capsids.

The DNA-filled capsid (Fig. 1a) has a diameter of 80 nm when measured from vertex to vertex, and extends 68 nm along the 2-fold symmetry axis. The most striking feature of the 7-7-1 capsid is the presence of capsid fibers, with one fiber emerging from the center of each vertex pentamer (Fig. 1b, left). However, due to the flexible nature of these capsid fibers, their densities average out during image processing. Therefore, we could only reconstruct a short section at the fiber base that connects to the capsid, and even these sections could only be resolved at a comparatively lower resolution. The capsid-fiber densities (Fig. 1b, green) were visualized in ChimeraX[24] using a lower contour level. Furthermore, the capsid belongs to a T = 9 ($h = 3$; $k = 0$; $T = h^2 + k^2 + hk$) triangulation symmetry with a planar outline. (Fig. 1c).

The capsid of phage 7-7-1 shows architectural similarity to a range of other phages, including *Ralstonia solanacearum* phage GP4[25], satellite phage P2[26], phage N4[27], *Anabaena* phage A-1(L)[28], *Helicobacter pylori* phages KHP30 and KHP40[29], and notably, phage Milano[30]. Although different in

## Table 1 | Cryo-EM data collection parameters and processing

| | Full capsid | Tail | Portal | Neck 1 | Neck 2 + tail terminator + tail sheath + tail tube | Collar | Baseplate core complex | Tail fibers |
|---|---|---|---|---|---|---|---|---|
| **Magnification** | 64,000 | | | | | | | |
| **Data collection/ Final Pixel size (Å)** | 0.685/1.37 | | | | | | | |
| **Voltage (kV)** | 300 | | | | | | | |
| **CS (mm)** | 2.7 | | | | | | | |
| **Detector** | Gatan K3 | | | | | | | |
| **Detector mode** | Super-resolution (non-CDS) | | | | | | | |
| **Energy filter (eV)** | 20 | | | | | | | |
| **Electron exposure (e-/Å2)** | 40 | | | | | | | |
| **Frames** | 40 | | | | | | | |
| **Defocus Range (µm)** | −1 to −2.5 | | | | | | | |
| **Number of movies collected** | 3537 | | | | | | | |
| **Initial particles (no.)** | 12,819 | 70,043 | 5771 | 5771 | 5771 | 5771 | 8266 | 48,966 |
| **Final particles (no.)** | 6975 | 21,858 | 4871 | 5578 | 4936 | 4792 | 8161 | 48,966 |
| **Symmetry** | I | C6 | C12 | C3 | C6 | C15 | C6 | C1 |
| **Helical rise (Å)** | - | 35.25 | - | - | - | - | - | - |
| **Helical twist (°)** | - | 27.35 | - | - | - | - | - | - |
| **Masked resolution at FSC = 0.143 (Å)** | 3.4 | 3.1 | 3.2 | 3.5 | 3.6 | 3.3 | 3.4 | 3.5 |
| **Masked resolution at FSC = 0.5 (Å)** | 3.8 | 3.3 | 3.6 | 4.1 | 4.1 | 3.7 | 4.0 | 4.0 |
| **Map sharpening B-factor (Å2)** | −116.7 | −111.7 | −61.5 | −35.0 | −39.6 | −63.4 | −48.5 | −45.9 |
| **EMDB** | EMD-52522 | EMD-52521 | EMD-54017 | EMD-54018 | EMD-54019 | EMD-54020 | EMD-54021 | EMD-54022 |

## Table 2 | Model refinement and validation statistics

| | Pentamer-hexamers | Tail | Portal | Neck 1 | Neck 2 + tail terminator + tail sheath + tail tube | Collar | Baseplate core complex | Tail fibers |
|---|---|---|---|---|---|---|---|---|
| **PDB** | 9HZ8 | 9HZ7 | 9RKQ | 9RKR | 9RKS | 9RKT | 9RKU | 9RKV |
| **Model compositions** | | | | | | | | |
| Amino Acids | 9694 | 7596 | 4680 | 1968 | 5592 | 12,480 | 18,903 | 4038 |
| **R.m.s. deviations** | | | | | | | | |
| Bond lengths (Å) | 0.003 | 0.003 | 0.012 | 0.004 | 0.006 | 0.003 | 0.002 | 0.003 |
| Bond angles (°) | 0.539 | 0.569 | 0.546 | 0.642 | 0.671 | 0.528 | 0.448 | 0.514 |
| **Validation** | | | | | | | | |
| MolProbity score | 1.33 | 1.43 | 1.08 | 1.70 | 1.85 | 1.03 | 1.48 | 2.01 |
| Clash score | 3.37 | 3.15 | 2.91 | 6.36 | 6.26 | 2.47 | 4.03 | 9.62 |
| Rotamers outliers (%) | 0.00 | 0.00 | 0.03 | 0.00 | 0.02 | 0.01 | 0.00 | 0.00 |
| **Ramachandran plot** | | | | | | | | |
| Favored (%) | 96.72 | 95.4 | 98.17 | 94.94 | 91.54 | 98.55 | 95.83 | 91.39 |
| Allowed (%) | 3.27 | 4.6 | 1.83 | 5.01 | 8.44 | 1.45 | 4.16 | 8.59 |
| Outliers (%) | 0.01 | 0.00 | 0.00 | 0.05 | 0.02 | 0.00 | 0.01 | 0.02 |
| **Rama-Z** | | | | | | | | |
| Whole | −0.56 (0.08) | −0.49 (0.10) | 1.18 (0.12) | 0.59 (0.21) | −1.10 (0.11) | 0.75 (0.07) | 0.01 (0.06) | −1.64 (0.13) |
| Helix | 1.18 (0.18) | 0.24 (0.14) | 1.81 (0.12) | 2.29 (0.21) | 0.73 (0.18) | --- (---) | 1.59 (0.08) | −0.01 (0.28) |
| Sheet | 0.21 (0.10) | 0.33 (0.12) | −0.25 (0.18) | −1.18 (0.25) | −0.42 (0.13) | 0.62 (0.06) | −0.48 (0.08) | −0.68 (0.20) |
| Loop | −0.95 (0.08) | −0.81 (0.09) | 0.28 (0.14) | −0.30 (0.23) | −1.33 (0.10) | 0.54 (0.08) | −0.63 (0.07) | −1.41 (0.11) |
| **ADP (B-factors)** | | | | | | | | |
| min/max/mean | 39.45/ 120.85/55.21 | 1.54/ 82.37/ 36.05 | 42.55/ 97.93/ 60.83 | 51.74/ 172.36/ 97.06 | 68.52/204.89/119.09 | 59.26/ 145.15/ 90.56 | 63.26/ 178.15/85.50 | 58.69/ 246.15/ 147.00 |
| **Model vs. Data** | | | | | | | | |
| CC (mask) | 0.84 | 0.85 | 0.83 | 0.73 | 0.82 | 0.83 | 0.85 | 0.81 |
| CC (volume) | 0.82 | 0.78 | 0.80 | 0.73 | 0.82 | 0.83 | 0.84 | 0.81 |

overall size, these phages share a common triangulation number of T = 9 as well as a similar HK97-fold capsid protein topology[31–34]. The 7-7-1 icosahedral capsid structure is constituted by a lattice of hexamers and pentamers, which are composed of a major capsid protein (MCP) (Fig. 1c). This capsid shell contains additional decoration proteins and linker proteins, which play an important role in enhancing capsid stability and potentially contribute to host recognition during the infection process. Specifically, the capsid of 7-7-1 consists of a total of 80 hexamers (Fig. 1a, blue) and 11 pentamers (Fig. 1a, cyan, with one vertex occupied by the portal complex), thus totaling 535 copies of the major capsid proteins. The MCPs and decoration proteins self-assemble to form the icosahedral shell that acts as the protective envelope for the phage genome.

### Hexamer and pentamer arrangements

The hexamer- and pentamer-forming major capsid protein (gp003, Gene ID: 14012067) of phage 7-7-1 (Fig. 2a) consists of 469 amino acids. Structurally, the MCP follows the classical HK97 fold, consisting of N-arm, A-domain, P-domain, G-loop, and E-loop (Fig. 2b, Supplementary Fig. 2a and b). The N-terminal residues M1 to A164 of the major capsid protein are not resolved, as they are cleaved during capsid maturation by a prohead protease[15]. The hexametric assembly unit of phage 7-7-1 is composed of six MCP monomers, forming a well-defined structure (Fig. 2a, blue). The pentamers are constituted by five MCPs (Fig. 2a, cyan). Compared to the hexamers, the pentamers adopt a slightly compressed configuration, accompanied by alterations in their relative orientation, resulting in a more curved arrangement within the pentamer. These structural differences are

most prominently seen in the N-arm and E-loop domains (Supplementary Fig. 2c and d). When comparing the hexamers surrounding the pentamer with those not in direct proximity to a pentamer, we observed a subtle augmentation in curvature. These observations provide new structural insights into the organizational dynamics of the capsid assembly of phage 7-7-1.

Hexamers and pentamers are further decorated with a decoration protein (DP, gp002, Gene ID: 14012068) and two linker proteins, namely linker protein 1 (LP1, gp117, Gene ID: 14012070) and linker protein 2 (LP2, gp122, Gene ID: 14012074). The decoration protein comprises 135 amino acids with a β-strand-rich domain (Fig. 2b). A dimer of decoration proteins embellishes the capsid surface by bridging the interfaces of pentamers and hexamers. The assemblies of MCP and DP complexes are further stabilized by linker protein 1 (LP1) and linker protein 2 (LP2), respectively. LP1 (gp117, Fig. 2b) on its own forms a relatively unstructured chain of amino acids, but interacts with the capsid facing interface of the DP and extends towards the center of the hexameric MCP complex (Fig. 2a). In contrast, LP2 (gp122 Fig. 2b) stabilizes the interaction between DPs and pentameric MCP assemblies (Fig. 2a). Furthermore, LP2 features a β-sheet Ig-type domain that is hypothesized to protrude the pentameric MCP complex into the poorly resolved capsid-fiber density, away from the capsid shell (Fig. 2c, green density, Supplementary Fig. 3). As the flexible density (marked in green) is not suitable for meaningful model building, we have used an AlphaFold prediction to depict the Ig-type domain of LP2 (Fig. 2b, encircled part of LP2). The predicted part is not included in the model shown in Fig. 2a. Ig-type domains are prevalent in phages, and their likely role

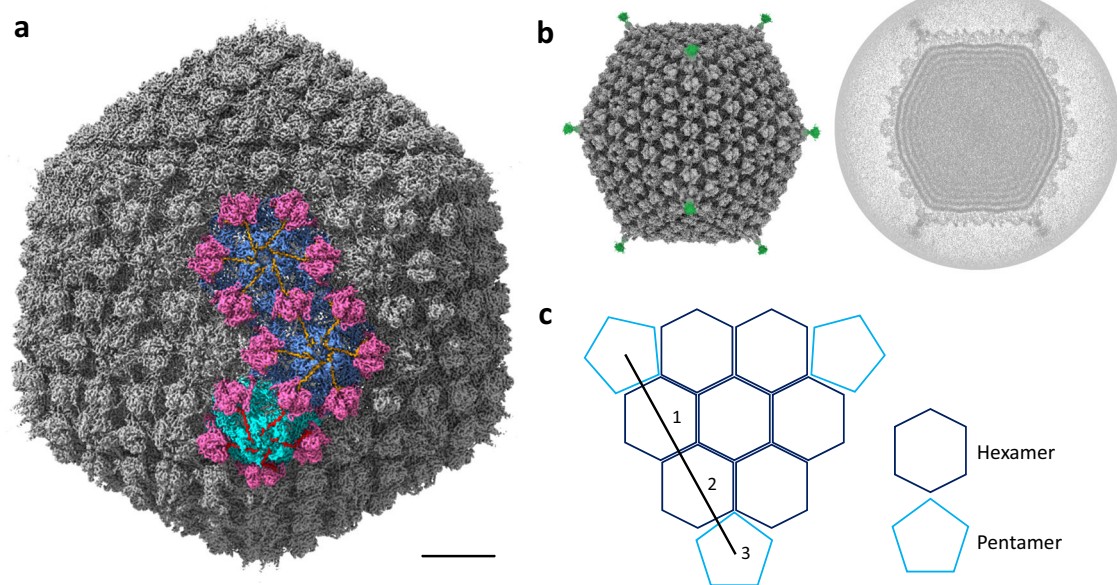

**Fig. 1 | Reconstruction and organization of the full DNA containing capsid of phage 7-7-1. a** The full icosahedral capsid of phage 7-7-1 filled with dsDNA was reconstructed at 3.4 Å resolution. Interlocking hexameric and pentameric capsid structures are highlighted in blue and cyan, respectively. Scale bar: 10 nm. **b** Left: the overall cryo-EM density map of phage 7-7-1, revealing the presence of a single capsid fiber at each vertex (green density). This representation used a low contour level. Right: a plane view of the dsDNA filled inside the capsid. **c** Schematic illustrating the 7-7-1 capsid organization, characterized by a T = 9 symmetry. The highlighted line in black signifies the *h* and *k* volumes involved in the calculation of triangulation symmetry[81].

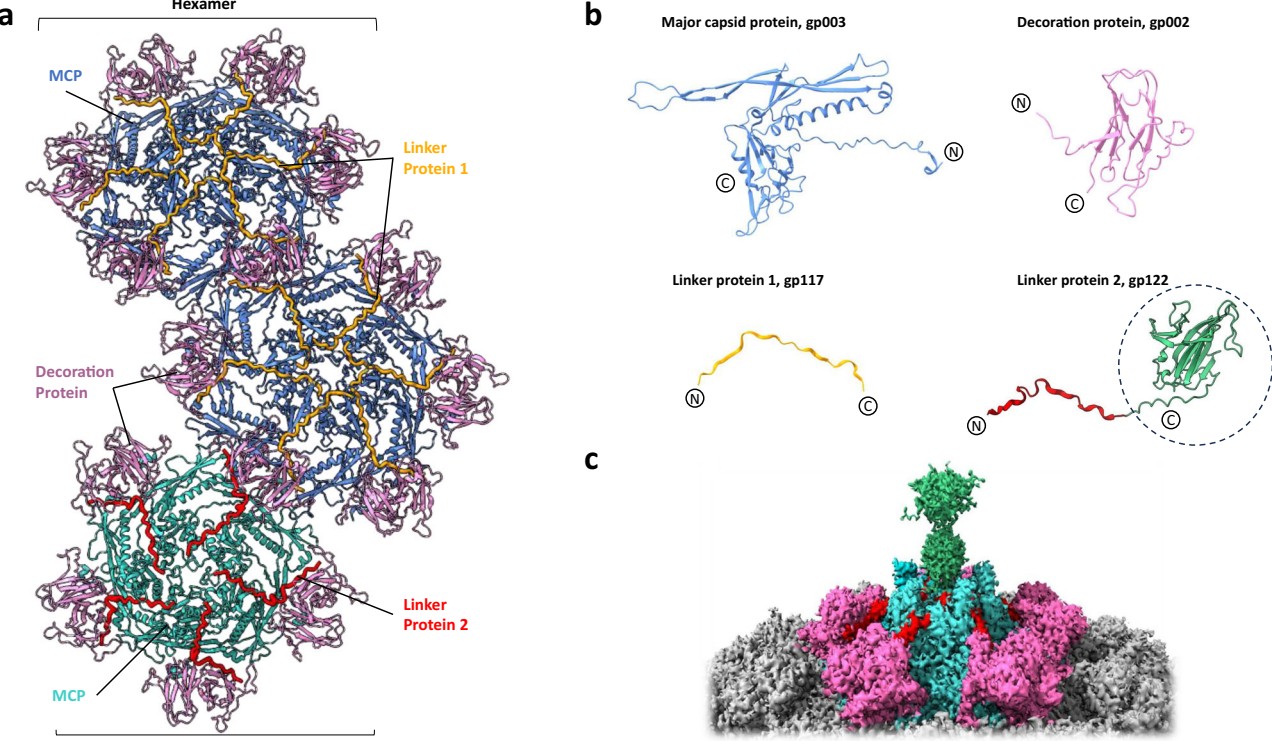

**Fig. 2 | Hexamer and pentamer arrangement in the phage 7-7-1 capsid. a** Structure of pentamer and hexamer differing in their decorating proteins (PDB: 9HZ8). MCP in hexamers are colored blue, MCP in pentamers are colored cyan. The decoration protein, linker protein 1, and linker protein 2 are colored pink, orange, and red, respectively. Each hexamer is decorated by six copies of linker protein 1, while each pentamer is decorated by five copies of linker protein 2. **b** Structures of MCP (gp003), decoration protein (gp002), linker protein 1 (gp122), and linker protein 2 (gp117) are shown with N-to C-terminal domains labelled. The Ig-type domain of linker protein 2 is predicted by AlphaFold, highlighted in green and circled, and not represented in the model in panel a. **c** The capsid-fiber density (green) protruding from the middle of a pentamer.

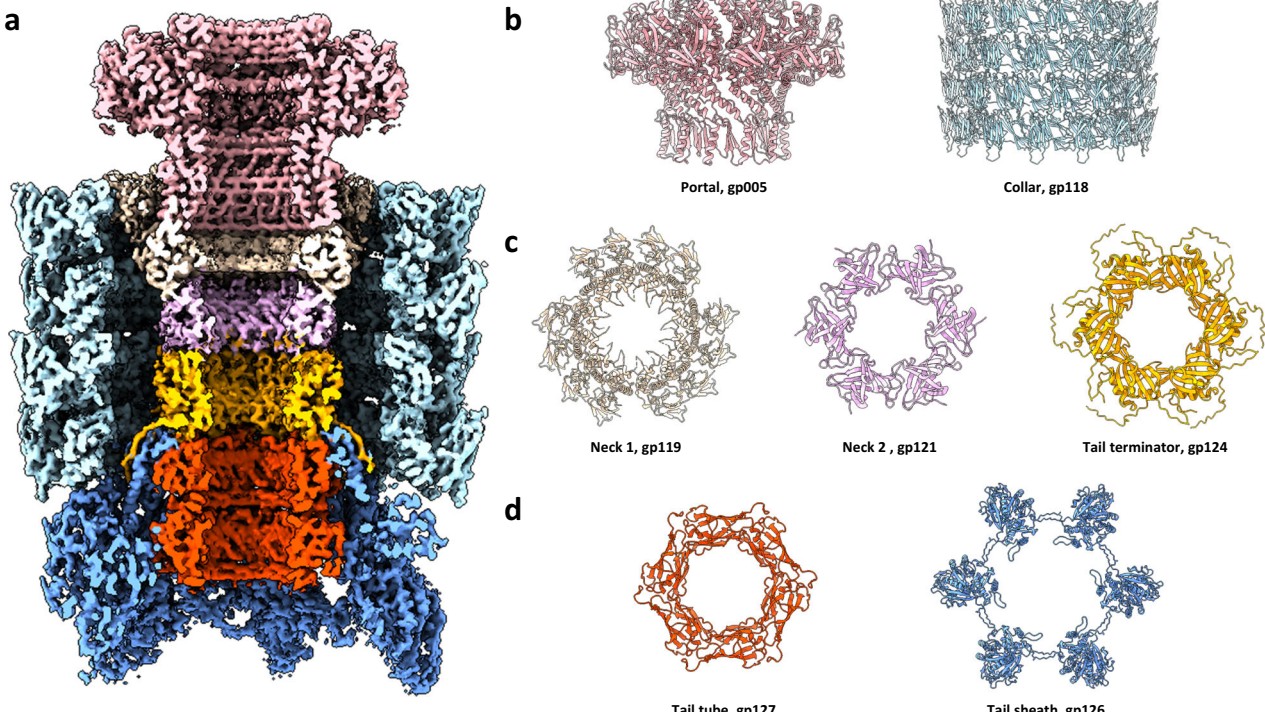

**Fig. 3 | Neck complex structure of phage 7-7-1. a** Cross section of the composite map of the neck region composed of and colored by the various neck region domains. Scale bar: 5 nm. **b** Structures of the portal (baby pink, gp005) and collar (light cyan, gp118) assemblies. **c** Structures of neck 1 (pale yellow, gp119), neck 2 (lavender, gp121), and the tail terminator (orange, gp124) regions. **d** Structures of the tail tube (red, gp127) and tail sheath (blue, gp126) components.

involves binding to cell surface components, thus tethering the phage to host cells[35–37]. We postulate that five β-sheet Ig-type domains of LP2 form a foundational structure at the center of the pentamer, serving as an initial anchoring point for the capsid fiber. As a whole, these four proteins make up the basic repetitive units that comprise the phage capsid.

Interestingly, there is a high abundance of cysteine residues in these four proteins (2.58% cysteine residues in total), all contributing to an interconnected disulfide bond network (Supplementary Fig. 4). In the hexameric conformation, the MCP has two internal disulfide connections (C190 + C389 and C302 + C460) and two linkages with neighboring MCPs (C444 + C227 and C186 + C459). Furthermore, it has two connections with the DPs (MCP C252 + DP C74 and MCP C223 + DP C104). The DPs share another two disulfide bonds with LP1 (DP C98 + LP1 C6 and DP C60 + LP1 C10). Lastly, LP1 makes an interconnected ring-like structure at the core of the hexameric assembly held together by interactions between C23 + C26. Together, these 9 unique disulfide bond classes contribute to a total of 54 disulfide linkages in the hexameric conformation. With a total of 35 disulfide connections, the pentameric conformation is formed in an almost identical fashion, but with only a single connection between DP and LP2 (DP C98 + LP1 C6), and there is no ring-like assembly at the core of the pentamer. As the unresolved Ig-type domain does not harbor any further cysteine residues, the mechanism of stabilization for these pentamer protruding parts remains unknown for now.

### Neck region of phage 7-7-1

Located at the junction between the capsid and the tail, the neck region of phage 7-7-1 serves as a structural and functional bridge between the head and tail components. Single particle analysis on this region revealed an intricate assembly hosting multiple symmetries. A cross-section of a composite map of the different locally refined neck region components discussed in this section is shown in Fig. 3. One of the pentameric vertices of the icosahedral capsid is replaced by a set of 12 portal proteins (portal, gp005, Gene ID: 14012065). The portal protein assembly initiates the inner lumen, which continues through the entire tail complex[38]. This channel facilitates

the transfer of viral genomic material from the capsid into the host cell during DNA injection. The portal was locally refined with C12 symmetry to a resolution of 3.2 Å and shares the portal fold organization commonly seen in phages (Fig. 3, baby pink).

The entire neck complex, from portal to tail region, is encased by a large external assembly called the phage collar. The collar is built from four stacked pentadecameric rings of the collar protein (gp118, Gene ID: 14011954). Together they form a 60-subunit scaffold that stabilizes the neck structure. Applying C15 symmetry, the collar was reconstructed to a resolution of 3.3 Å (Fig. 3, light cyan).

The collar and portal are linked via neck 1 (gp119, Gene ID: 14012071), which also forms a dodecameric ring of proteins (Fig. 3, pale yellow) and was reconstructed at a resolution of 3.5 Å using C3 symmetry. The region of neck 1, situated directly below the portal assembly, abides C12 symmetry. Interestingly, the portion of neck 1 directly interacting with the collar breaks C12 symmetry and serves as a converter between the mismatching C12-C15 symmetries of the collar and portal, respectively. Each of the 12 neck 1 subunits align between two adjacent collar subunits. Approximately after every four neck 1 subunits, one position remains unoccupied, effectively converting C15 symmetry to C12. Applying C1 reconstruction symmetry, we do see weak neck 1 electron densities in the skipped positions, indicating that the arrangement of the neck 1 subunits might be somewhat flexible and can adopt multiple conformations.

The C12 symmetry of the portal and inner parts of the neck 1 ring is reduced to C6 by a hexameric ring of neck 2 protein (gp121, Gene ID: 14012073) (Fig. 3, lavender). Each neck 2 protein is positioned between two copies of neck 1. The sixfold symmetry is maintained throughout most of the tail structure, including the sheath and major parts of the baseplate. Neck 2 also interacts with a hexameric ring of tail terminator proteins (TTP, gp124, Gene ID: 14012076) (Fig. 3, orange). A long loop domain of neck 2 (residues 14–30) resides at the interface of two neighboring TTPs (residues 106–140 and 32–40), resulting in a stable hexamer-hexamer interface. Interestingly, the C-terminal domain of TTP (residues 160–176) extends outward as a flexible arm that contacts the first layer of sheath protein

**Fig. 4 | Tail structure arrangement in phage 7-7-1.**
**a** Top: Side view of 7-7-1 tail density with two sheath subunits rows highlighted. Bottom: Top view of the 7-7-1 tail structure, the outer sheath and inner tube colored with a gradient of blue-green and orange-pink, respectively (scale bar: 5 nm). **b** Left: Ribbon diagram of the tail complex model. Sheath subunits are highlighted. Right: The ribbon model of 7-7-1 sheath protein gp126. Interaction domains highlighted in green (residues 1-25 and 385-500). **c** Left: Organization of the tail tube complex. Right: The ribbon model of 7-7-1 tube protein gp127. The interprotein interaction domains, N-terminus (residues 1-7), C-terminus (residues 128-135), and loop (residues 43-61) are highlighted in magenta.

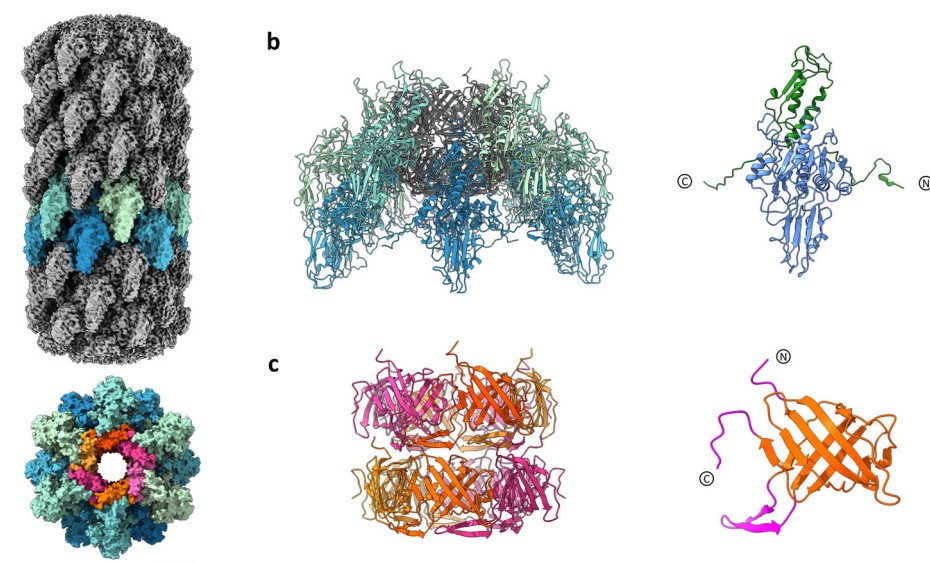

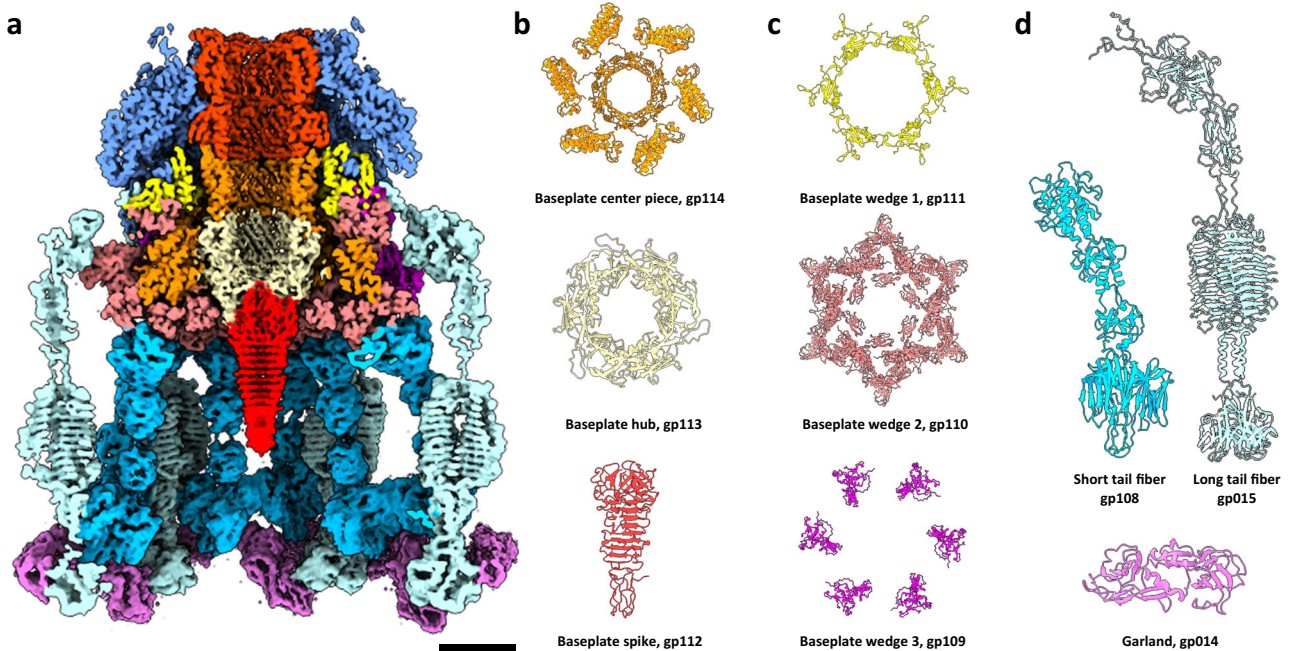

**Fig. 5 | Baseplate complex structure of phage 7-7-1. a** Cross-sectional representation of the composite map from the end of the tail region (top: tail tube: red, tail sheath: blue) into the baseplate complex, color-coded by individual domains. Scale bar: 5 nm. **b** Models of the baseplate center piece (orange, gp114), baseplate hub (pale yellow, gp113), and baseplate spike (bright red, gp112) forming the baseplate inner hub. **c** Structures of baseplate wedge 1 (yellow, gp111), baseplate wedge 2 (pink, gp110), and baseplate wedge 3 (magenta, gp109). **d** Structures of the short tail fiber (sky blue, gp108), long tail fiber (light cyan, gp015), and garland proteins (candy pink, gp014).

(gp126, Gene ID: 14012078) (Fig. 3, blue), reinforcing the neck-to-sheath transition. The first hexameric ring of the tail tube (gp127, Gene ID: 14012079) (Fig. 3, red) defines the end of the neck region and the beginning of the phage tail region. The N-terminal region of the tail tube protein (residues 1–7) extends upwards to form a connection with TTP (residues 10–12 and 126–128). In addition, extruding loops from both the tail tube and terminator proteins form top-down contacts at residues 31–37 and 71–77, respectively. The assembly of neck 2, TTP, tail tube, and tail sheath was locally refined to a resolution of 3.6 Å. Together, the portal, collar, neck 1, neck 2, and the tail terminator proteins form an intricate symmetry bridging assembly that links the icosahedral capsid to the contractile tail machinery, which will be discussed in detail in the following sections.

## Tail structure of phage 7-7-1

The tail of phage 7-7-1 consists of two stacked components: the outer sheath (Fig. 4b, gp126, Gene ID: 14012078) provides the mechanical force during injection, while a hollow inner tube (Fig. 4c, gp127, Gene ID: 14012079) forms a channel for the transfer of genetic cargo[39]. SPA analysis of the tail of phage 7-7-1 revealed its similarity to those of other structurally described *Myoviridae* (Fig. 4a, resolution: 3.0 Å). Based on individual micrographs and this density map, we observed that the extended tail is 135 nm in length and has a diameter of about 20 nm.

The stacked sheath proteins form a C6-symmetric helical interlocking structure. Every 7-7-1 sheath subunit has two long arm extensions at the N- and C-termini. Accompanied by two neighboring subunits, each arm

contributes to the formation of an interlocking β-sheet motif (Fig. 4b). This in turn leads to a helical assembly of the tail sheath complex (twist of 27.35° and a rise of 35.25 Å). Furthermore, the tail sheath complex has a lumen of approximately 70 Å in which the inner tube resides. The 7-7-1 inner tube structure also exhibits a C6 symmetry, aligning with the surrounding sheath proteins (Fig. 4c). Tube protein subunits oligomerize into hexameric ring-like structures that stack on top of one another, creating a hollow β-barrel tube structure with an inner diameter of approximately 40 Å. The three main interaction domains that stabilize the tube assembly (the N-terminal, C-terminal, and loop interaction domain) are highlighted in pink. Together, the tail sheath and tube proteins form a robust conduit, guiding the genetic payload from the capsid to the baseplate.

### Baseplate structure of phage 7-7-1

The baseplate complex terminates the tail apparatus at the distal end and plays a crucial role in host recognition, irreversible binding, and initiating sheath contraction to facilitate genome delivery into the host[40]. The baseplate follows a common organizational theme observed in a variety of contractile-tailed phages[41]. The central inner hub is surrounded by baseplate wedges forming the core of the baseplate complex, which in turn are decorated by tail fibers (Supplementary Fig. 1A, purple arrows).

The inner hub acts as the transition site from the repeating tail tube to the central spike protein and is composed of the baseplate center piece protein (gp114, Gene ID: 14011956), the baseplate central hub protein (gp113, Gene ID: 14011957), and the central spike protein (gp112, Gene ID: 14011958). The baseplate center piece forms a hexameric ring and resides directly below the last tail tube protein ring (Fig. 5, orange). The N-terminal region (residues 1–127) closely resembles the β-barrel fold characteristic of the tail tube and connects to the tail tube above through its protruding N-terminal part (residues 1–10) and the baseplate hub below (residues 31–56). The C-terminal region (residues 128–398) in the baseplate center piece consists of alpha helical bundles that make up the central scaffold of the baseplate wedge complex, which will be discussed later. By linking these two structural regions, the baseplate center piece serves as a bridge between the baseplate central hub and the surrounding wedge complex. In turn, the baseplate hub trimer reduces the predominant C6 symmetry down to C3 symmetry to accommodate the baseplate central spike (Fig. 5, pale yellow). The baseplate central spike is a trimer that folds into a triple β-helix domain, characteristic of spike proteins in Myoviruses[42] (Fig. 5, bright red). The baseplate central spike seals the tail lumen and acts as the battering point of the bacteriophage that penetrates the host cell's outer membrane[43].

The inner hub structure of the baseplate is encased by wedge proteins, forming an interlocking mesh with the sheath proteins, central hub, and tail fiber proteins. The baseplate wedge complex comprises wedge 1 (gp111, Gene ID: 14011959), wedge 2 (gp110, Gene ID: 14011960), and wedge 3 (gp109, Gene ID: 14011961). These structures intimately interact with the C-terminal domain of the baseplate center piece, making up the characteristic volume of the baseplate. Wedge 1 forms a hexameric ring and functions as a transition element between the last row of tail sheath proteins and the baseplate assembly (Fig. 5, yellow). The protruding N-terminal domain of the tail sheath protein is captured by a distinctive pocket in wedge 1, which otherwise represents the binding surface for the neighboring sheath subunit. Six dimers of wedge 2 also form a hexameric assembly and harbor globular domains that project inwards toward the lumen of the baseplate and interact with the central hub of the baseplate, functioning as a latch to hold the inner structure in place (Fig. 5, pink)[44]. Surrounding the wedge 2 ring, six copies of wedge 3 complete the baseplate outer architecture and, together with wedge 1 and 2, facilitate the binding of the short and long tail fibers (Fig. 5, magenta).

Attached to the baseplate are six long tail fibers and twelve short tail fibers. Phage tail fibers are crucial for host recognition and attachment during bacteriophage infection. They act as receptor-binding proteins, specifically interacting with molecules on the bacterial cell surface to initiate the infection process[45].

The long tail fiber is composed of three subunits (gp015, Gene ID: 14012055) (Fig. 5, light cyan). The N-terminal region (residues 1–120) mediates anchoring interactions with the core of the baseplate, the middle section forms a beta-helical domain (residues 177–453), and the C-terminal part is made up of a triple jelly roll domain (residues 473–587). The overall architecture of the long tail fiber trimer is consistent with other characterized long tail fiber trimers[46,47].

The short tail fiber is a smaller trimeric structure consisting of four domains (gp008, Gene ID: 14012062). Each long tail fiber is accompanied by two short tail fibers, resulting in a total of twelve short tail fiber trimers per phage (Fig. 5, sky blue). The N-terminal region of the trimer forms a six alpha helix bundle and interacts with wedge 2 and wedge 3 (residues 1–160). The middle two regions consist of smaller globular domains (residues 82–161). The C-terminal region is formed by a triple jelly roll motif and a smaller protruding spike-like domain (residues 162–298), which has been proposed to bind to the host surface and potentially have enzymatic properties[45]. This region interacts with neighboring short tail fibers, the long tail fibers, and the garland proteins.

The most distal part of the *Agrobacterium* phage 7-7-1 baseplate region is surrounded by a garland made up from a total of 36 homodimeric garland proteins (gp014, Gene ID: 14012056). The garland proteins interact with the lower region (middle and C-terminal domains) of both short and long tail fiber complexes. Due to the small size of the individual proteins and seemingly flexible nature of the garland as a whole, only the most rigid garland dimer pairs could be confidently resolved and modeled (Fig. 5, candy pink). The garland has been observed in phages T4 and Milano[48,49], and is involved in irreversible binding of the baseplate to the host surface. As a whole, the intricate design of the baseplate makes up a finely tuned molecular machine that can initiate infection through host binding and a variety of structural transitions.

### Comparison of phage 7-7-1 structural proteins with phages Milano, E217 and T4

Phage 7-7-1, together with phages Milano, E217, and T4, all belong to the contractile-tailed *Myoviridae*. Due to their shared morphology and infection strategy, we compared the structures resolved here with those of these well-characterized references. To assess these relationships, we performed structural alignments between the structural proteins of phage 7-7-1 and the respective references using RMSD and TM-score (Supplementary Fig. 5 and Supplementary Table 1).

Among these, phage Milano exhibits the highest degree of structural conservation with 7-7-1. Most proteins show TM-scores above 0.9 and low RMSD values (<1.5 Å), indicating nearly identical folds. In contrast, E217 shares moderate structural similarity with phage 7-7-1, with TM-scores ranging between 0.4 and 0.6 for most components. Selected core elements like the capsid, portal, tail tube, and baseplate hub, retain significant structural similarity (TM-scores 0.6−0.7), but others show substantial divergence. Comparison with phage T4 reveals the greatest divergence, with most components exhibiting TM-scores smaller than 0.6 and higher RMSD values spanning over the shortest regions of similarity. Even conserved structural elements like the tail tube and portal show TM-scores around 0.5, suggesting that 7-7-1 and T4 are evolutionarily and architecturally more distinct. These findings underscore that phage 7-7-1 shares a conserved structural framework with other Myoviridae-like phages. Especially components like the major capsid protein, the portal, the contractile tail, and the baseplate hub seem highly conserved, while other regions like the neck and peripheral host-recognition components show greater divergence or are even absent from the assembly and functionally organized in a different manner.

### Cryo-electron tomography and segmentation of phage 7-7-1

Phage 7-7-1 was further investigated by cryo-ET to investigate the structurally flexible components of phage 7-7-1, as well as to gain insight into

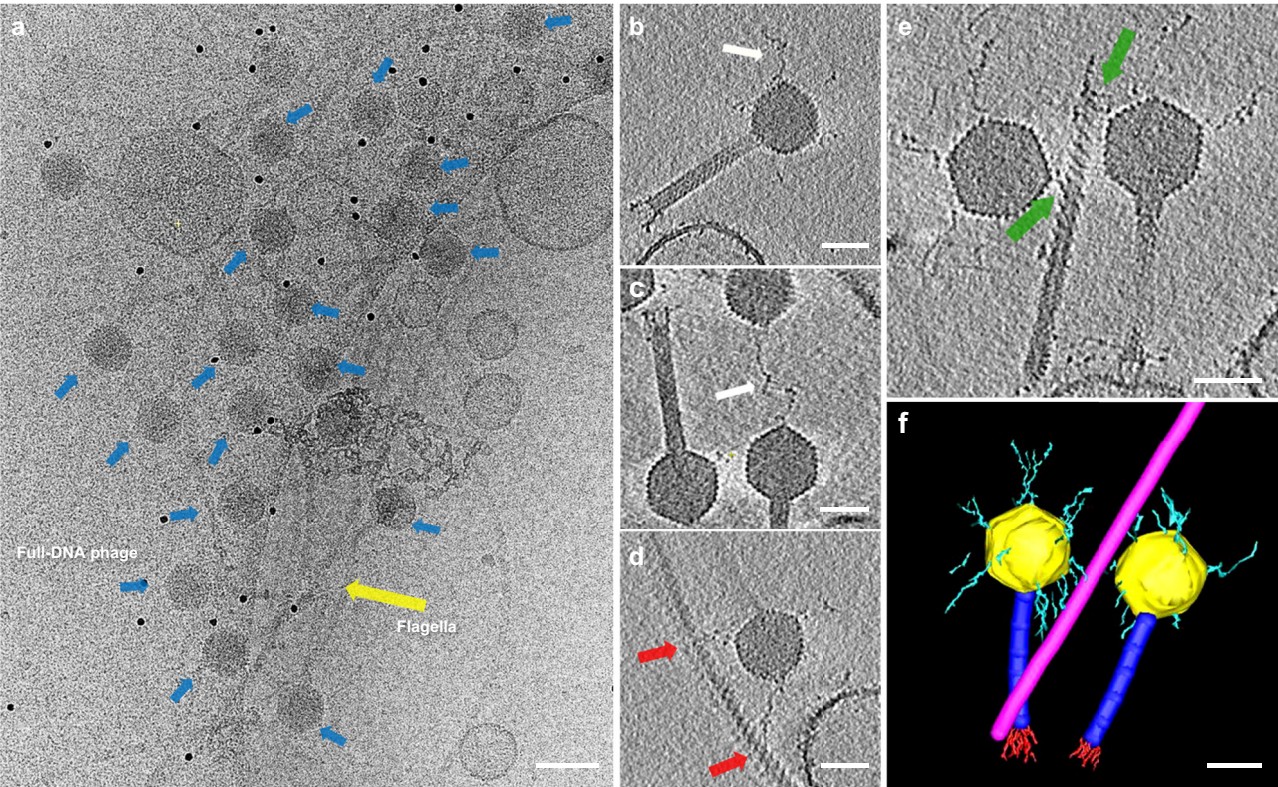

**Fig. 6 | Cryo-electron tomography and segmentation of phage 7-7-1 attached to** *Agrobacterium* **flagellum. a** A representative cryo-ET cross-section (slice) is presented, revealing multiple instances of full-DNA phage 7-7-1 (blue arrows) attached to a cluster of flagella (yellow arrow). The tomographic data clearly reveal the presence of the characteristic icosahedral capsid, contractile tail, tail fibers, and capsid fibers. **b** Example of long flexible fibers (white arrow) of a full-capsid phage 7-7-1, originating from the capsid vertices. **c** Occasionally, it can be observed that the flexible capsid fibers are connecting two phages (white arrow). **d** The interaction of two phage capsid fibers with the host flagellum (red arrows). **e** Two phages attached to the same flagellum (green arrows), further emphasizing the strong interaction between phage 7-7-1 and the host flagellum. **f** A volumetric representation of the tomogram subregion shown in **e**, with manual segmentation performed using IMOD. Scale bars: **a**: 100 nm, **b–f**: 50 nm.

host-phage interactions during the initial infection process. On many occasions, we found an accumulation of phage particles in close proximity to a cluster of flagella (Fig. 6a)

Notably, the capsids of phage 7-7-1 were oriented toward the flagella, indicating their capability to establish connections with the flagella via structures located on their capsids. The accumulation of full-DNA phages around the flagella provided evidence of initial phage-host interaction of 7-7-1. Furthermore, analysis of individual phages in the tomograms revealed that the capsids, while in close proximity, appear not to be in direct contact with the flagella. This suggests that the interaction of the phage with the bacterial flagellum involves the fibrous filaments located on the capsid.

To investigate the architecture and function of the flexible capsid fibers during the various stages of the infection process, we focused on complete phage 7-7-1 particles interacting with its host. More specifically, we explored phage 7-7-1 in conjunction with its host flagellum, leading to the identification of four distinct states: free particles with and without DNA and adsorbed particles with and without DNA. These selected particles represent the different stages during phage infection, including free phages (state 1), phages during the early-infection stage once attached to the host (state 2), post-infection (empty) phages still attached to the cell (state 3), and post-infection (empty) free phages (state 4).

The pre-infection phages (state 2) filled with DNA were used to study the interaction between capsid fibers and flagella. Sub-volumes of individual full-capsid phages were extracted from the tomograms. They confirmed the presence of long, flexible fibers connecting the capsids with the flagellar filament (Fig. 6b–e). More specifically, one long, flexible fiber emerges from each of the vertices of the 7-7-1 capsid. The combination of our SPA and tomography results has revealed that the fibers emerge from

the 11 pentamers on the capsid. However, due to the missing wedge artifact in the tomograms, which is caused by the limited tilt range of the specimen holder and blocks full sample illumination and the inherent flexibility of the fibers, not all capsid fibers were fully visible in the tomograms and the SPA reconstruction. Furthermore, accurate measurements of the fiber length posed challenges due to their flexible and curly nature.

Intriguingly, we found that the capsid fibers of phage 7-7-1 not only seem to interact with the host's flagellum but could also interact with fibers from other 7-7-1 capsids (Fig. 6c, white arrow, Supplementary Fig. 6). However, the nature of the interaction between the fibers and their potential role to aid each other's probability of success for infection remains unknown. At present, the proteins, their functions, and the interaction mechanisms of the head fibers with host flagella are also unknown. Previous studies have identified three candidate receptor binding proteins (RBPs), gp4, gp102, and gp44, based on their ability to bind to host cells[14]. It remains to be seen whether these three putative RPBs specifically interact with host flagella. Figure 6d further highlights the intriguing observation of one flagellum attached to two 7-7-1 phages by capsid fibers, suggesting the possibility of multiple phages concurrently attaching to the same flagellum via multiple capsid fibers. A closer inspection of the helical flagellar architecture depicted in Fig. 6c–e reveals that the phage capsid fibers wrap around the flagellum. This observation corresponds to the established model describing the interaction between the capsid fiber of bacteriophage φCbK and flagella of its host[50]. Proteinase K treatment was utilized to determine if the capsid fibers of bacteriophage 7-7-1 are formed by protein. Cryo-EM imaging of the micrographs revealed a progressive reduction in the number of capsid fibers with increased incubation time (Supplementary Fig. 7), indicating the

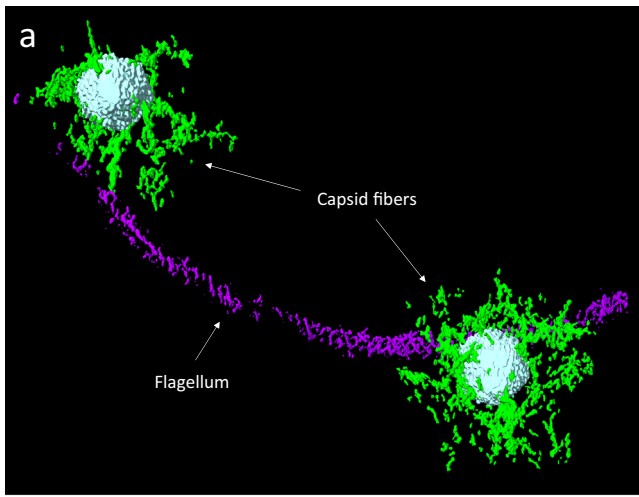

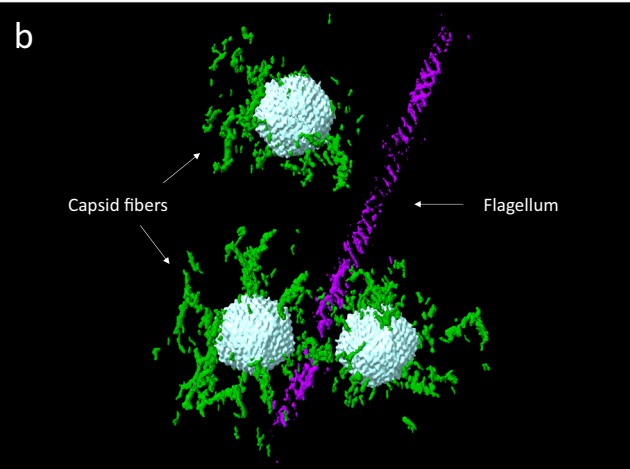

**Fig. 7 | 3D structures of capsid fibers of phage 7-7-1 attached with flagellum.** Two examples (**a**, **b**) of 3D structures of the capsid (cyan), capsid fibers (green), and flagellum (purple) that were generated using the neural network. Flagellum-associated phage 7-7-1 particles are oriented with their capsids toward and close to the flagellum. Structures of phage 7-7-1 with capsid fibers extending from the phage and wrapping around the flagellum.

time-dependent degradation of the capsid fibers upon Proteinase K treatment.

Due to the intricate complexity and heterogeneous nature of the capsid fibers, manual segmentation using the IMOD segmentation function[51] was chosen for initial analysis. Consequently, the sub-tomogram presented in Fig. 6d was manually segmented in IMOD and revealed multiple capsid fibers attached to the flagellum (Fig. 6f). These findings shed light on the complex interactions between phage 7-7-1 and its host's flagella, underscoring the importance of such connections in the infection process.

**Capsid fiber analysis using machine learning**

To streamline the labor-intensive process of manually segmenting capsid fibers, we applied our previously developed neural network to automate the detection of tail fibers in reconstructed tomograms of phage φKp24[52]. Here, this network was specifically trained using manual segmentations of capsids, capsid fibers, and flagella, and subsequently optimized through multiple iterations. Once trained, the network was applied to all tomograms, resulting in the generation of comprehensive 3D maps that accurately captured the structures of capsid fibers and flagella. This approach significantly reduced the time and effort required for fiber detection and facilitated efficient analysis of the phage structures. These results illustrate the structural attachment of phages via their capsid filaments to the flagellum during the initial infection stage.

As mentioned above, we defined the pre-infection phages based on four key characteristics: (1) the presence of a full capsid, (2) attachment of capsid fibers to flagella, (3) intactness of the phage structure, and (4) absence of contact with a bacterial cell. After extraction and format conversion, 13 capsid fiber structures of pre-infection phages were generated. Through visual inspection, we carefully selected and presented 6 groups of representative trimmed tomograms and their corresponding capsid fiber structures in Supplementary Fig. 8. These 3D structures exhibit distinct conformations for each capsid fiber, highlighting the dynamic nature of these components.

Following an extraction and format conversion process, we successfully generated 13 pre-infection phage 7-7-1 structures that were attached to the flagellum. Measuring the length of capsid fibers within these structures revealed a range between ~ 88 nm to 114 nm. While the flexibility and curvature of capsid fibers posed challenges to obtain precise measurements, these findings demonstrate the remarkable capability of phage 7-7-1 to establish a robust attachment, but at a considerable distance between the capsid and the flagellum.

In Fig. 7a, b, we present two representative 3D structures of capsid fibers from phage 7-7-1 attached to the flagellum. These 3D structures exhibit distinct conformations for each capsid fiber, highlighting their dynamic nature. Particularly, the depicted structures in Fig. 7b illustrate the capsid fibers extending from the phage and enveloping the flagellum. This mode of fiber-flagellum interaction closely resembles the well-established interaction observed between the capsid fiber of bacteriophage φCbK and its host flagellum[50].

## Discussion

Capsid fibers assume a critical role in facilitating the attachment of viral capsids to host receptors. Various phages and viruses exhibit intriguing variations in the structural characteristics of these capsid fibers. For instance, *Bacillus subtilis* phage φ29[53] contains capsid fibers that are attached to quasi-3-fold symmetry positions on the capsid, characterized by a protruding stem without a terminal sphere[22,54]. Other phages, such as *Caulobacter crescentus* phage φCbK and *Vibrio alginolyticus* phages φSt2 and φGrnl feature only a solitary flexible fiber positioned atop the viral head[50,55,56]. Some *Vibrio* phages possess fibers, often referred to as antennae, terminated by trilobate structures anchored on the viral head[57]. Another form of capsid fibers can be found in the *Agrobacterium tumefaciens* phage Atu_ph07, which resembles thin hair-like fibers[58]. Here, we determined a novel arrangement of capsid fibers in phage 7-7-1, with one fiber emerging from each vertex. More specifically, the fibers emerge from each pentamer of the capsid, totaling 11 capsid fibers. The final (12th) vertex is occupied by the phage-tail connection.

Despite the new structural insights into the unusual capsid and capsid fibers of phage 7-7-1 reported here, many questions remain. For example, the structure of the 7-7-1 capsid fiber is still unknown, and we can only suggest that the five β-sheet Ig-type domains of link protein 1 gp122 form a foundational structure at the center of the pentamer, serving as an initial anchoring point for the capsid. Drawing an analogy to the fiber proteins found in human adenoviruses, which are also positioned at each vertex of the capsid, we postulate that 7-7-1 capsid fibers might share a common architectural framework. According to this model[59], the capsid fiber protein can be envisioned as a trimer, with each monomer comprising an N-terminal tail, a central shaft consisting of repetitive sequences, and a C-terminal globular knob domain. In this context, the N-terminal ~45 residues of the fiber protein are notably conserved across various phages and are primarily responsible for binding to the penton base[60,61]. On a structural level, within one capsid fiber, a fiber protein trimer potentially binds to a pentameric complex located at the fivefold symmetry axes of the capsid. For an individual monomer within the trimeric protein, approximately 11 residues of the N-terminal tail form stable contacts by inserting between two adjacent penton monomers[62], which means there is a stable attachment between the fiber and the virion when other regions of the fiber move[63,64]. By featuring 11 capsid fibers, phage 7-7-1 circumvents steric limitations presented by the relatively flat capsid surface. This architectural adaptation

likely enhances the phage's ability to bind to flagella in random orientations and at different rotational speeds, which may improve the likelihood and efficiency of attachment.

Deciphering the unique morphology of the 7-7-1 capsid presents a notable challenge for structural investigations. The considerable size of the capsid and the heterogeneity of the capsid fibers introduce complexities in both data collection and analysis. They require the integration of diverse complementary structural techniques to achieve a comprehensive understanding of the overall structure.

While previous work has only indicated the presence of capsid head fibers in low-resolution EM images, we gained more detailed insight into the structural importance of head fibers in the infection process of phage 7-7-1. Due to the challenging, flexible nature of these fibers, little structural and functional insight was available. Here we used cryo-ET to image the host and phage 7-7-1 in 3D and at macromolecular resolution. Initial imaging revealed that the capsids of phage 7-7-1 were oriented toward flagella but at a considerable distance. This already suggests their ability to establish direct connections with the flagella via fibers situated on their capsid surfaces. Furthermore, our data showed that multiple phages can attach to a single flagellum, which could indicate a collaborative association potentially facilitated by the flagella themselves.

Surprisingly, we found that phage 7-7-1 can also form connections with other phages via their head fibers. We propose that phage 7-7-1 could act collaboratively to accomplish successful infection: an initial phage seeks out and attaches to a flagellum, and subsequently, another phage can locate and connect to the first phage via the capsid fibers, thus facilitating its proximity to the bacterial cell and subsequent completion of infection. The interaction details between head fibers are still unknown and need further investigation in the future. In addition, we observe the adsorption of multiple capsid fibers to the flagellum, illustrating the robust attachment capability of the 7-7-1 capsid to flagella.

Drawing an analogy to the capsid fibers observed in phages φCb13 and φCbK[50], which are also positioned at each vertex of the capsid, we hypothesize that the capsid fibers of 7-7-1 may share a common structure and employ a similar interaction mechanism with the flagella of the host cell. It is plausible that phage 7-7-1 initially adheres to the flagellum through the head filament, subsequently leading to an irreversible attachment of the phage tail to the lipopolysaccharide (LPS) layer of the host, which serves as a secondary cell surface receptor. Consequently, we surmise that the interaction between the head filament of phage 7-7-1 and the flagellum resembles that observed between the φCbK phage's head fiber and the flagellum[55,65]. In this model, a portion of the phage's head fiber encircles the flagellum, facilitating the localization of bacteriophages to the cell surface as the flagellar filament rotates. Subsequently, the tail of phage 7-7-1 is anticipated to bind to the LPS culminating in irreversible attachment and DNA injection[20].

## Materials and methods
### Sample preparation
*Agrobacterium* sp. H13-3 was grown in NY at 30°C to an $OD_{600}$ of 0.6, diluted into 200 mL NY to an $OD_{600}$ of 0.03, infected with phage at an MOI of 0.005, and incubated with shaking at 30 °C for 24 h. Sodium chloride was added to a final concentration of 4%, left on ice for 30 min, and centrifuged at $10,000 \times g$ for 30 min at 4 °C. Precipitation was accomplished by adding polyethylene glycol 8000 to the supernatant (10% w/v) and incubation at 4 °C for 16 h. Phage particles were sedimented by centrifugation, suspended in 2 mL of TM buffer (10 mM MgSO4 and 20 mM Tris/HCl, pH 7.5), and overlaid on a 10% to 50% (w/v) linear iodixanol (OptiPrep; Accurate Chemical and Scientific Corporation, Westbury, NY, USA) gradient. Following centrifugation at $200,000 \times g$ for 2 h at 15 °C using an SW-41 Ti rotor, a blue-white band containing virions was removed with an 18-gauge syringe and dialyzed against TM buffer at 4 °C with two buffer changes. Following this protocol, phage titers ranged from $10^{11}$-$10^{12}$ plaque-forming units (pfu) mL$^{-1}$. The final phage stock was stored in TM buffer at 4 °C[20,66].

Phage 7-7-1 was further concentrated as followed: 1000 μL of phage 7-7-1 preparation ($1\times10^{12}$ pfu/mL) was added to a Centricon with a 10-kDa

cellulose filter and centrifuged 2-3 times at $5000 \times g$ for 15 min each to approximately 75 μL until no buffer eluted from the protein concentrator.

For SPA reconstruction, the samples were plunge frozen using the Leica EM GP: 3.0 μL of the concentrated phage was added to a glow-discharged Quantifoil R2/2, 200 mesh Cu grid, incubated at 20 °C with approximately 85% relative humidity, pre-blotting time was 30 s, blotted for 0.7 s, and automatically plunged into liquid ethane.

### Imaging conditions for SPA
Phage 7-7-1-containing grids were clipped and loaded into a Titan Krios electron microscope (Thermo Fisher Scientific) operated at 300 kV, equipped with a K3 direct electron detector operating in super-resolution mode and BioQuantum energy filter using a slit width of 20 eV (Gatan, Inc). Micrographs were collected using the SerialEM data collection software[67], and locations containing multiple phages were manually selected for imaging. A total of 3537 movies were recorded at a nominal magnification of 64.000, corresponding to a calibrated super-resolution pixel size of 0.685 Å (physical pixel size 1.37 Å), with a defocus range of −1 to −2.5 μm in steps of 0.25 μm, and a total dose of 40 e/Å$^2$ spread over 40 frames (see data collection parameters in Table 1).

### SPA data processing
Data processing was performed using CryoSPARC[68]. Micrographs were imported, patch motion corrected, and patch CTF estimated. Particles were manually picked for template generation and subsequent template picking. The tail regions were picked using filament tracing. After multiple rounds of 2D classifications, the best classes were submitted to ab-initio model generation using various symmetries depending on the structure of interest (capsid: I, portal: C12, collar: C15, neck 1: C3/C1, neck 2 + tail terminator protein + tail tube + tail sheath: C6, tail tube + tail sheath: C6 helical, baseplate C6/C3, baseplate tail fibers: C1). The resulting best model was used for homogeneous and/or non-uniform refinements, with local and global CTF corrections, as well as Ewald sphere correction. For the helical refinement of the tail region, the corresponding initial model was analyzed with the symmetry search utility and yielded an estimated 27° twist and a 35 Å rise, which was used in subsequent refinements. After refinement, the particles were submitted to reference-based motion correction and a final round of refinement as described before, yielding the final reconstructions. Reconstructed volumes containing multiple symmetries (neck region and baseplate region) were then individually masked and locally refined. Map-map Fourier shell correlation curves of all maps (cut-off 0.143) are shown in Supplementary Fig. 9.

### Model building
ModelAngelo[69] was used for initial model generation without providing sequence input. Subsequently, protein sequences were identified with findMySequence[70] referenced against the partially annotated phage 7-7-1 genome[17] and fed back into ModelAngelo. The sequence-based model was then analyzed in ChimeraX[24], and any missing regions were added with the Modeller package[71]. The models were then interactively improved using ISOLDE[72] and refined in Phenix[73]. The model-building statistics are available in Table 2. Map-model Fourier shell correlation curves of all maps (cut-off 0.143) are shown in Supplementary Fig. 10.

### Sample preparation for cryo-ET
*Agrobacterium* sp. H13-3 strain was grown overnight in TYC (0.5% tryptone, 0.3% yeast extract, and 0.087% CaCl$_2$ x 2H$_2$O [pH 7.0]) at 30 °C in a shaking incubator at 180 rpm/min. Two hundred microliters of bacterial cell cultures ($OD_{600}$ = 0.2) were sedimented at $3000 \times g$ for 15 min at room temperature and suspended in 15 μl motility buffer (0.5 mM CaCl$_2$, 0.1 mM EDTA, and 20 mM HEPES, pH 7.4). Subsequently, the concentrated cell suspension was mixed with 15 μl of phage 7-7-1 ($1\times10^{12}$ pfu/mL) and 2 μl 10-nm gold beads (Cell Microscopy

Core, Utrecht University, Utrecht, The Netherlands). The mixture was incubated at room temperature without shaking for 3–10 min prior to plunge freezing. Using the Leica EM GP (Leica Microsystems, Wetzlar, Germany), 3.8 μL of the bacteria, phage and gold bead solution (volume ratio: 15:15:2) was applied to a glow discharged Quantifoil R2/2, 200 mesh Cu grid (Quantifoil Micro Tools GmbH, Jena, Germany), which was incubated for 30 s prior to blotting at 20 °C with approximately 95% relative humidity. The grids were blotted for 1 s and automatically plunged into liquid ethane. Vitrified samples were transferred to storage boxes and stored in liquid nitrogen until use.

### Imaging conditions for cryo-ET
The grids containing *Agrobacterium* sp. H13-3 bacteria and phage 7-7-1 were clipped and loaded into a Titan Krios (Thermo Fisher Scientific (TFS)) transmission electron microscope equipped with a K3 Bio-Quantum (Gatan, Inc) direct electron detector operating in counting mode and the energy filter set to a 20 eV slit. Targets were chosen manually based on the presence of flagella attached to bacterial cells that were located in a hole of the carbon film of the EM grid. A total of 60 tilt-series of phage 7-7-1 attached to flagella were collected using SerialEM set to a dose symmetric tilt scheme between −54° and 54°, with 2° tilt increments[67,74]. The selected nominal magnification was 26,000, which corresponds to a pixel size of 3.28 Å. The defocus of the collected data ranged from −4 to −6 μm with 0.5 μm increments. The total dose per tilt series was 100 e-/Å$^2$, corresponding to 1.82 e-/Å$^2$ per tilt.

### Cryo-ET data processing
Motion correction, dose-weighting, per-tilt CTF correction, tilt series alignment using 10 nm gold fiducials, and tomogram reconstruction were carried out using the IMOD software package[51], and IsoNet was used to restore isotropic resolution and reduce missing wedge artifacts prior to segmentation.[75] The tomograms were binned by a factor of 2. From the reconstructed tomograms, regions showing phages interacting with flagella were selected for segmentation using IMOD. Visualization of data was performed in IMOD and Fiji[76].

### Proteinase K treatment
Two microliters of Proteinase K (10% w/v) were added to 18 μl of the concentrated phage 7-7-1 suspension and incubated at 37 °C for 20, 40, and 60 min. For subsequent imaging, 3 μl of the mixture was added to a glow-discharged Quantifoil R2/2, 200 mesh Cu grid, incubated for 10 s at 20 °C with approximately 85% relative humidity, pre-blotting time was 30 s, blotted for 0.7 s, and automatically plunged into liquid ethane. All the vitrified samples were stored in liquid nitrogen until use.

### Fiber analysis using machine learning
A 100-layer mixed-scale dense neural network was trained to detect the capsid fibers of phage 7-7-1 in a reconstructed tomogram. For the training process, manual segmentations of the tail fibers from 9 phages were utilized[77–79]. The ADAM algorithm[80] was employed to minimize the cross-entropy loss, with random rotations and flips applied for data augmentation. The entire training process took a few hours and was halted when no significant improvement in the loss was observed. After training, the network was applied to all tomograms, yielding 3D maps of the fiber structures. To perform additional analyses, manual annotation was conducted on the tomograms for each phage, identifying the center of the fiber structure and the position of the head. Using this annotated information, a 3D map of the fiber structure for each phage was extracted from the 3D fiber maps, resulting in individual maps for 13 capsid fiber structures of pre-infection phage. To ensure proper alignment, the maps were aligned to each other using both the manually annotated position information and subsequent automatic alignment through maximizing cross-correlation. The extracted structures were visualized using ChimeraX for further examination and analysis.

### Reporting summary
Further information on research design is available in the Nature Portfolio Reporting Summary linked to this article.

### Data availability
The 3D cryo-EM maps generated in this study have been deposited at EMDB (the Electron Microscopy Data Bank, https://www.ebi.ac.uk/emdb/) with accession code EMD-52522 for the full capsid, code EMD-52521 for the tail, code EMD-54017 for the portal, code EMD-54018 for neck 1, code EMD-54019 for Neck 2 + tail terminator + tail sheath + tail tube, code EMD-54020 for the collar, code EMD-54021 for the baseplate core complex, and code EMD-54022 for the tail fibers. Representative cryo-electron tomograms have been deposited at EMDB under accession code EMD-53352. Raw tomography data have been submitted to EMPIAR under accession code EMPIAR-13014. Protein prediction data reported in this paper will be shared by the lead contact upon request. The atomic coordinates generated in this study have been deposited at wwPDB (Worldwide Protein Data Bank, http://www.wwpdb.org/) with accession PDB ID: 9HZ8 for pentamer-hexamers of the full capsid, accession PDB ID: 9HZ7 for the tail (2 layers), accession PDB ID: 9RKQ for the portal, accession PDB ID: 9RKR for neck 1, accession PDB ID: 9RKS for Neck 2 + tail terminator + tail sheath + tail tube, accession PDB ID: 9RKT for the collar, accession PDB ID: 9RKU for the baseplate core complex, and accession PDB ID: 9RKV for the tail fibers. Source data for Supplementary Figs. 9 and 10 are in the Supplementary Data.

### Code availability
All original code has been deposited at GitHub and is publicly available via https://github.com/dmpelt/phage771code and https://zenodo.org/records/17581374. Any additional information required to re-analyse the data reported in this paper is available from the lead contact upon request.

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

## Acknowledgements
This work benefited from access to the Netherlands Centre for Electron Nanoscopy (NeCEN) at Leiden University, which was funded in part by the Netherlands Electron Microscopy Infrastructure (NEMI), project number 184.034.014 of the National Roadmap for Large-Scale Research Infrastructure of the Dutch Research Council (NWO). This work was funded by the National Science Foundation fund number IOS-2054392 to BES. R.O. was supported by the China Scholarship Council (CSC) with project number 201906280465. We thank Floricel Gonzalez and Abigail Horton for 7-7-1 phage preparations and Frank Aylward and Abdeali Jivaji for help with the ANI analysis. We thank Miguel Leung for discussions on SPA data processing.

## Author contributions
Conceptualization: A.B.; methodology: A.B., A.S., N.C.E., R.O., T.H., W.N.; formal analysis: A.S., N.C.E., R.O., T.H., W.N.; investigation: A.S., N.C.E., R.O., T.H., W.N.; resources: A.B., B.E.S., D.M.P.; data curation: A.B., A.S., B.E.S., N.C.E., R.O., T.H., W.N.; writing original draft: A.B., A.S., R.O., T.H., W.N.; writing—review and editing: all authors; supervision: A.B.; project administration: A.B.; funding acquisition: A.B., A.S., B.ES., D.M.P., R.O.

## Competing interests
The authors declare no competing interests
