## [Transparent Peer Review file · Communications Biology]

Insights into the structure and initial host attachment of the flagellotropic bacteriophage 7-7-1

Corresponding Author: Professor Ariane Briegel

Version 0:

Reviewer comments:

Reviewer #1

(Remarks to the Author)

This is a well-written manuscript that advances the field of phage research by providing novel insights into flagellotropic phages, which have not been studied extensively. In particular, the authors use an innovative combination of cryo-EM SPA and ET methods in combination with machine learning, enabling them to track and segment the novel flexible capsid fibers, and directly visualizing the interaction between capsid and flagella, and also between capsids. Furthermore, they clearly demonstrate the time-dependent fiber degradation with a proteinase-K assay, which correlates well with infectivity estimated in a spot assay. The phage itself is a new member among only few examples of jumbo phages, which adds to the significance of this work.

The structure and role of viral capsid fibers is still not fully understood, and more research is necessary. This work pushes the frontier to the nature of flagellotropic interactions. The tomography is excellent, as expected from this group. Although the resolution of the individual tomograms is limited due to the missing wedge, and subtomogram averaging is challenging due to the flexibility of the fibers, the gained insights describe a new paradigm how phages interact with flagella. I find the author's hypothesis about a collaborative phage network in proximity to flagella refreshing and stimulating. The paper will therefore appeal to a broad audience and is well suited for publication in Comms Bio.

Minor comments/suggestions:

- Fig.1 legend and also main text – “small threshold level of 0.01 in chimerax”: this value is not very informative, unless the reader is using the same software. The authors could normalize the map (average 0, stand dev 1) and specify the contour level as multiple of standard deviations.
- If the genes for the fibers are known, they could be compared by sequence alignment with others such as in ϕ Cb13 and ϕ CbK. Could AlphaFold3 produce a meaningful structure?
- The tomograms should also be deposited to the EMDB

Reviewer #2

(Remarks to the Author)

In this manuscript entitled “Using cryogenic electron microscopy methods to gain insight into the structure and initial host attachment of the flagellotropic bacteriophage 7-7-1”, Noteborn et al. determined the atomic structures of capsid and tail from bacteriophage 7-7-1 by cryo-EM. They also investigated the interactions between 7-7-1 and host flagellum by cryo-ET and Proteinase K treatment analysis.

All the results give us an insight into the architecture of bacteriophage 7-7-1 and describe the interactions involving capsid fiber and tail fiber. This study is solid due to its experimental design and mythology in the cryo-EM field. The experimental design is generally coherent, the methods are proper, and the results are meaningful. The more impressive part is that they use machine learning to pick up the phages and show different types of interactions.

The main suggestions for this study are as follows:

1. Page 12, Line 284-285: "Our SPA result has revealed that the fibers emerge from the 11 pentamers on the capsid. However..."

The reconstruction of capsid relies on I symmetry (Supplementary Table 1), which means the densities of 12 pentamers have been averaged, including the "capsid fiber" (root part on Capsid). Thus, although it's highly possible that each pentamer vertex has a capsid fiber (except the tail vertex), the SPA structure alone can not prove that point. Unless there is other evidence.

2. In Figure 1a and Figure 3a, adding a length marker could give the reader a direct feeling about the size.

3. In Figure 6a, "no treatment," "20 min," "40 min," and "60 min" could be added in the left corner of each image, which will be much clearer.

4. In Figure 6b, why the "20 min" treatment data is missing?

5. Page 19, Line 468, "xx ul," xx should be a number.

6. Although it's a supplementary figure, Supplementary Figure 1 still needs scale bars in the figure.

Reviewer #3

(Remarks to the Author)

Summary:

Noteborn et al. use cryo-electron microscopy, machine-learning, and molecular biology techniques to investigate the structural basis of host cell attachment by bacteriophage 7-7-1. The approach is similar to the authors' previous work (Ouyang et al., 2022), but is nonetheless distinct as here the methods are used to study capsid fibers rather than tail fibers. The results here will certainly be of interest to readers studying viruses using similar mechanisms of host-attachment, as well as to structural virologists for the methods employed by the authors.

Nevertheless, I have questions and suggestions for the authors to consider that I hope will add clarity to the manuscript prior to its publication.

Questions:

• Ln70-71: Are the authors able to add specifics to the statements in these lines?

What is the percent identity between 7-7-1 and Milano/OLIVR4 bacteriophages?

What is considered an "unusual accumulation of cysteine residues"? What is the percentage of cysteine residues in 7-7-1 structural proteins relative to other Myoviridae of similar genome size?

• Ln119-122: Can the authors prepare a main-text figure panel and/or supplementary figure showing an enlarged view of the reconstructed capsid fiber density? The capsid fibers are a major point in the manuscript and such a figure would help readers appreciate how difficult they are to interpret from the icosahedral averaged reconstruction.

• Ln136-140: In addition to the same T-number, do all the phages share a similar HK97-fold topology and capsid size?

• Ln162-164: What is the percent identity and RMSD between the 7-7-1 MCP and the canonical HK97-fold? Can the authors also provide a main text figure panel or SI figure with the 7-7-1 MCP labelled according to the conventional HK97-fold nomenclature (i.e., A-domain, P-domain, E-loop, etc.)? This can be added SI Figure 2, as it will help to highlight the major difference of hexamer and pentamer protomers conformations in the E-loop.

• Ln166-168: Are the residues unresolved because of high flexibility or is the MCP expressed in a pro-form that is matured through cleavage by a prohead protease like in other phages? Does this phage encode a prohead protease? Is there proteomics or SDS-PAGE of virions to suggest proteolysis of the MCP in the mature virion?

• Ln180-182: What is the percent sequence identity and RMSD between the additional proteins of 7-7-1 and Milano?

• Ln184-187: Can the authors provide a figure or panel showing the disulfide bonds between the linker and MCPs? Are there examples of such covalent links in other known phage capsids?

• Ln193-200 and Figure 2: Is the Ig-like domain of gp122 resolved in the map? Its location in the coordinate model of Figure 2a is unclear and the domain is colored grey instead of rainbow in 2b. Can 2c be labelled to emphasize the location of the Ig-like domain?

- Are the authors able to reveal any additional structural information of the capsid-fiber base from either C1 reconstructions of the capsids or from symmetry-expansion methods such as those outlined here:

<https://guide.cryosparc.com/processing-data/tutorials-and-case-studies/case-study-end-to-end-processing-of-encapsulated-ferritin-empiar-10716?>

The capsid fibers are a major focus of this manuscript and any additional structural detail(s) that could lead to the components identification would be impactful.

- I highly recommend the authors deposit the data on EMPIAR for general data transparency, as well as encouraging methods development on difficult targets such as the capsid fibers presented in the manuscript.
- The T=9 capsid asymmetric unit has 8 hexameric MCP protomers and 1 pentameric MCP protomer, but the presented model consists of 12 hexameric MCP and 5 pentameric MCP protomers. Can the authors provide a rationale for modelling the components as presented?
- Have the coordinate models for the capsid and tail been refined with the same symmetry relation among chains as enforced on the reconstruction?
- Can the authors please add the map-vs-model FSC for the modelled components to SI figure 5, as well as add the FSC0.5 and refined ADPs values to the SI table?

Suggestions for clarity

- Ln26-27: remove “,and built atomic models of capsid hexamers, pentamers and tail” since it reads redundantly to “determined capsid and tail structure”
- Ln30-31: I suggest adding context to this sentence such as “time-depended degradation of capsid fibers [correlated with reduced infectivity]” or something to that effect.
- Ln55-57: I recommend omitting this sentence since it adds specifics on Agrobacter that are not further discussed in the manuscript.
- Figure 1:
 - o a: It would be helpful to have the symmetry axes indicated on the map to help orient the reader on such a large structure.
 - o a: Please indicate the length of the scale bar on the figure or in the legend.
 - o The Caspar-Klug system (Caspar and Klug, 1962) might be a more appropriate reference than viralzone.expasy.org for triangulation number.
- Ln136-140: Milano is referenced twice as a close relative. Omit one of the references
- Ln349: remove ‘=’ after “nature”
- Ln468: replace “xx” with value.

References (in order of appearance):

Ouyang R, Costa AR, Cassidy CK, Otwinowska A, Williams VCJ, Latka A, Stansfeld PJ, Drulis-Kawa Z, Briers Y, Pelt DM, Brouns SJJ, Briegel A. 2022. High-resolution reconstruction of a Jumbo-bacteriophage infecting capsulated bacteria using hyperbranched tail fibers. *Nat Commun* 13:7241.

Caspar DL, Klug A. 1962. Physical principles in the construction of regular viruses. *Cold Spring Harb Symp Quant Biol* 27:1–24.

Reviewer #4

(Remarks to the Author)

In the paper titled, Using cryogenic electron microscopy methods to gain insight into structure and initial host attachment of the flagellotropic bacteriophage 7-7-1, by Noteborn et al., the authors used cryo-electron microscopy, cryo-electron

tomography and machine learning to examine phage-host interactions. They were able to visualize capsid fiber binding around the bacterial flagellum at higher resolutions than have previously been described. The authors also generated atomic models of the phage capsid and tail.

There are no page or line numbers, but the authors state around page 12, Intriguingly, we found that the capsid fibers of phage 7-7-1 not only seem to interact with the host's flagellum but could also interact with fibers from other 7-7-1 capsids (Figure 4c, white arrow). I could not find anywhere that it was stated how many times the authors made this observation? I am wondering if this could also have happened by chance because the phage heads were near each other rather than showing an actual interaction. Is there any evidence that these capsid fibers interact with each other in other systems? Could the proteins that are predicted to exist on the termini be overexpressed and then examined directly? The reason I wonder if this is a real observation or an artifact is because these fibers would need to be expressed within the bacterial cytoplasm and if they bind to each other, it is difficult to imagine how they assemble with the correct stoichiometry at each annex and do not agglutinate.

For Figure 6, I would argue that it is difficult to conclude that this assay is indicating the impact of the capsid fiber on bacterial infectivity. I do not think the authors have any evidence that the capsid fibers are more sensitive to PK digestion than the tail fibers that are also important for host binding. Also, most of the phages shown after 60 min incubation with PK appear to look just like the empty capsid phages observed in Suppl Fig 1b and so these phages would also be unable to infect their host cells.

In the Materials and Methods section in the first paragraph, a number is missing in the following sentence: centrifuged 2-3 times at 5,000 × g for 15 min each to approximately xx µl until no buffer eluted from the protein concentrator.

Reviewer #5

(Remarks to the Author)

The authors present structural studies of bacteriophage 7-7-1, a phage that initiates infection by attaching to host flagella. Cryo-EM is employed to reconstruct the phage's capsid and tail structures at high resolution, drawing insight into the arrangements of pentameric and hexameric subunits, and enabling atomic modeling of their constituent proteins. To investigate capsid fibers, which are hypothesized to play a critical role in the initiation of infection, cryo-ET is additionally employed to visualize phages in the context of their flagellar interactions. Based on machine learning-guided segmentation of phage structures within tomograms, the authors posit that capsid fibers form the initial attachment between the phage and a flagellum, and may additionally be involved in the recruitment of other phage particles.

The manuscript provides structural insights into the plant bacteria-infecting bacteriophage 7-7-1, adding to, and providing additional validation for, structural studies conducted on other flagellotropic phages. The use of cryo-ET in conjunction with machine learning-based segmentation is a promising approach suggesting that capsid fibers play a critical role in phage attachment and recruitment. However, the employed machine learning architecture and training strategy, which the authors had developed in a previous publication, appear to produce capsid fiber segmentations that are suboptimal in quality. Improved prediction or visualization of these segmentations may clarify the evidence the authors present in favor of the functional hypotheses they propose.

Comments:

1) Compared to the manual capsid fiber segmentation presented in Figure 4, the automated segmentations presented in Figures 5 and S4 appear to be of noticeably lower quality. The fragmented nature of these automated segmentations make their visual interpretation challenging, especially in relation to observations of "capsid fibers extending from the phage and enveloping the flagellum." Could the quality of the visualizations be improved by noise removal or manual cleanup? Alternatively, by performing cross-validation on your set of manual annotations, could you employ common evaluation metrics for cryo-ET segmentation such as the DICE and Surface-DICE scores discussed in Lamm et al. [1], in order to assess the accuracy of your trained model?

2) An interesting hypothesis is presented that "an initial phage seeks out and attaches to a flagellum, and subsequently, another phage can locate and connect to the first phage via the capsid fibers, thus facilitating its proximity to the bacterial cell and subsequent completion of infection". Since you have comprehensively segmented tomograms, you might consider highlighting these connections in a figure to provide further evidence.

3) Is there a reason that the phage tails are omitted from the training of the automated segmentation model? The annotation of tails in Figure 4 is helpful in visualizing phage orientation, and could be similarly helpful in Figure 5.

[1] Lamm et al. MemBrain v2: an end-to-end tool for the analysis of membranes in cryo-electron tomography. biorXiv 2024.

Version 3:

Reviewer comments:

Reviewer #1

(Remarks to the Author)

The authors have addressed all my comments.
Congratulations to another solid bacteriophage paper.

Reviewer #2

(Remarks to the Author)

The authors have addressed all my concerns in the revised manuscript, and it is now ready for publication.

Reviewer #3

(Remarks to the Author)

The authors have either met or exceeded my expectation in their revisions and I applaud their addition of the neck and tail-tip machinery reconstructions to the manuscript! I endorse the manuscript for publication.

Reviewer #4

(Remarks to the Author)

The authors have addressed all the reviewers comments effectively. This manuscript provides new insights into flagellotropic phages and will be of interest to many readers.

Reviewer #5

(Remarks to the Author)

I appreciate the authors' clarification of the challenges in segmentation, their use of IsoNet to enhance tomogram interpretability, and the inclusion of a new SI figure highlighting the connections between phages. I believe the machine learning analysis supports their novel structural insights and have no further concerns.

List of Responses

Dear Editors and Reviewers:

Thank you for your constructive comments concerning our manuscript entitled " Using cryogenic electron microscopy methods to gain insight into structure and initial host attachment of the flagellotropic bacteriophage 7-7-1". We have addressed each input as outlined below, which has improved the manuscript significantly. The revised text is marked red in the manuscript to highlight the changes we've made. The main corrections in the manuscript and the responses to the reviewers' comments are outlined below.

Responds to the reviewers' comments:

Reviewer 1:

This is a well-written manuscript that advances the field of phage research by providing novel insights into flagellotropic phages, which have not been studied extensively. In particular, the authors use an innovative combination of cryo-EM SPA and ET methods in combination with machine learning, enabling them to track and segment the novel flexible capsid fibers, and directly visualizing the interaction between capsid and flagella, and also between capsids. Furthermore, they clearly demonstrate the time-dependent fiber degradation with a proteinase-K assay, which correlates well with infectivity estimated in a spot assay. The phage itself is a new member among only few examples of jumbo phages, which adds to the significance of this work.

The structure and role of viral capsid fibers is still not fully understood, and more research is necessary. This work pushes the frontier to the nature of flagellotropic

interactions. The tomography is excellent, as expected from this group. Although the resolution of the individual tomograms is limited due to the missing wedge, and subtomogram averaging is challenging due to the flexibility of the fibers, the gained insights describe a new paradigm how phages interact with flagella. I find the author's hypothesis about a collaborative phage network in proximity to flagella refreshing and stimulating. The paper will therefore appeal to a broad audience and is well suited for publication in Comms Bio.

We thank the reviewer for the kind remarks about our manuscript and for the useful comments, which we address in detail below.

Minor comments:

1. Fig.1 legend and also main text – “small threshold level of 0.01 in chimerax”: this value is not very informative, unless the reader is using the same software. The authors could normalize the map (average 0, stand dev 1) and specify the contour level as multiple of standard deviations.

Response:

We agree with the reviewers' comment. We have adapted the wording in the text for explaining the difference in contour levels. In line with other reviewers' comments to make this display the capsid fibers better, we have adapted figure 2C, where we have better represented the capsid fiber, and have made a SI figure where we show enlarged side and top views of the fiber density.

2. If the genes for the fibers are known, they could be compared by sequence alignment with others such as in ϕ Cb13 and ϕ CbK. Could AlphaFold3 produce a meaningful structure?

Response: Unfortunately, the 7-7-1 genes are unknown to us and we could not find any capsid fiber sequence similarity between other species (including ϕ Cb13 and ϕ CbK). In a brute force attempt, we even predicted the entire proteome with AF3 and checked for candidate structures and interactions, but none were identified.

3. The tomograms should also be deposited to the EMDB.

Response: Two representative cryo-tomograms have been deposited to the EMDB under number EMD-53352. The main text has been modified to include this reference.

Reviewer 2:

In this manuscript entitled “Using cryogenic electron microscopy methods to gain insight into the structure and initial host attachment of the flagellotropic bacteriophage 7-7-1”, Noteborn et al. determined the atomic structures of capsid and tail from bacteriophage 7-7-1 by cryo-EM. They also investigated the interactions between 7-7-1 and host flagellum by cryo-ET and Proteinase K treatment analysis.

All the results give us an insight into the architecture of bacteriophage 7-7-1 and describe the interactions involving capsid fiber and tail fiber. This study is solid due to its experimental design and methodology in the cryo-EM field. The experimental design is generally coherent, the methods are proper, and the results are meaningful. The more impressive part is that they use machine learning to pick up the phages and show different types of interactions.

We thank the reviewer for the positive comments on our manuscript and for the constructive criticism provided, which we address in detail below.

Main suggestions:

1. Page 12, Line 284-285: “Our SPA result has revealed that the fibers emerge from the 11 pentamers on the capsid. However...” The reconstruction of capsid relies on I symmetry (Supplementary Table 1), which means the densities of 12 pentamers have been averaged, including the “capsid fiber” (root part on Capsid). Thus, although it's highly possible that each pentamer vertex has a capsid fiber (except the tail vertex), the SPA structure alone can not prove that point. Unless there is other evidence.

Response: Thank you for pointing this out- this indeed needs to be further elaborated. Indeed, SPA alone would not prove the presence of a capsid fiber at each pentamer vertex given the I symmetry. However, the combination of SPA and tomography supports this conclusion: In the tomograms we can clearly see fibers emerging from all pentamers (unless the density is obscured by the ‘missing wedge’, the imaging artifact that originates due to the incomplete tilt range during data collection). We have adapted the text accordingly.

2. In Figure 1a and Figure 3a, adding a length marker could give the reader a direct feeling about the size.

Response: Thanks for the suggestion. We have added scale bars with numbers to both figures.

3. In Figure 6a, “no treatment,” “20 min,” “40 min,” and “60 min” could be added in the left corner of each image, which will be much clearer.

Response: Thanks for the suggestion. We have adapted the figure accordingly.

4. In Figure 6b, why the “20 min” treatment data is missing?

Response: After internal consideration we have decided to remove the panel on infectivity from the manuscript as it was not conclusive enough and did not highly contribute to the main text. We do thank the reviewer for their comment.

5. Page 19, Line 468, “xx ul,” xx should be a number.

Response: Thanks for pointing this out. We have added the correct number of microliters (75 ul).

6. Although it’s a supplementary figure, Supplementary Figure 1 still needs scale bars in the figure.

Response: Thank you for pointing this out. We have added scale bars to this figure.

Reviewer 3:

Noteborn et al. use cryo-electron microscopy, machine-learning, and molecular biology techniques to investigate the structural basis of host cell attachment by bacteriophage 7-7-1. The approach is similar to the authors’ previous work (Ouyang et al., 2022), but is nonetheless distinct as here the methods are used to study capsid fibers rather than tail fibers. The results here will certainly be of interest to readers studying viruses using similar mechanisms of host-attachment, as well as to structural virologists for the methods employed by the authors.

Nevertheless, I have questions and suggestions for the authors to consider that I hope will add clarity to the manuscript prior to its publication.

We thank the reviewer for the kind remarks about our manuscript and for the useful

comments, which we address in detail below.

Comments:

1. Ln70-71: Are the authors able to add specifics to the statements in these lines?

What is the percent identity between 7-7-1 and Milano/OLIVR4 bacteriophages?

What is considered an “unusual accumulation of cysteine residues”? What is the percentage of cysteine residues in 7-7-1 structural proteins relative to other Myoviridae of similar genome size?

Response: We thank the reviewer for this comment and agree more elaboration is necessary. We have added the specifics on the percent identity between the different species (80% in both cases) and have gone deeper into the importance of the cysteine residue accumulation (1.45% whole genome, 2.8% in the structural proteins studied here). This will also follow in answer to comment #7.

2. Ln119-122: Can the authors prepare a main-text figure panel and/or supplementary figure showing an enlarged view of the reconstructed capsid fiber density? The capsid fibers are a major point in the manuscript and such a figure would help readers appreciate how difficult they are to interpret from the icosahedral averaged reconstruction.

Response: Thank you for this comment. We agree with the reviewer that the capsid fiber density was too poorly represented. We have adapted figure 2 and made a new supplementary figure 3 to better represent the capsid fiber density.

3. Ln136-140: In addition to the same T-number, do all the phages share a similar HK97-fold topology and capsid size?

Response: Thanks for the reviewer's question. According to the referenced studies, in addition to sharing the same $T = 9$ number, the major capsid proteins of all the listed phages from Line 136 (*Ralstonia solanacearum* phage GP4[26], satellite phage P2[27], phage N4[28], *Anabaena* phage A-1(L)[29], *Helicobacter pylori* phages KHP30 and KHP40[30], and phage Milano[31]) exhibit a similar HK97-fold topology. However, they do not share identical capsid sizes. For example, the diameter of phage P2 is approximately 60 nm, phage GP4 is 74 nm, phage KHP30 is 70 nm, and our phage 7-7-1 is about 80 nm. We have adapted the main text to reflect these statements.

4. Ln162-164: What is the percent identity and RMSD between the 7-7-1 MCP and the canonical HK97-fold? Can the authors also provide a main text figure panel or SI figure with the 7-7-1 MCP labelled according to the conventional HK97-fold nomenclature (i.e., A-domain, P-domain, E-loop, etc.)? This can be added SI Figure 2, as it will help to highlight the major difference of hexamer and pentamer protomers conformations in the E-loop.

Response: We thank the reviewer for this comment that will indeed clarify the structural features of the MCP. We have added the HK97 fold labelling and the comparison with the classical HK97 fold and the RMSD between the two structures to Supplementary Figure 2. We have also discussed the differences in the N-arm and E-loop between the hexamer and pentamer conformation in the main text.

5. Ln166-168: Are the residues unresolved because of high flexibility or is the MCP expressed in a pro-form that is matured through cleavage by a prohead protease like in other phages? Does this phage encode a prohead protease? Is there proteomics or SDS-PAGE of virions to suggest proteolysis of the MCP in the mature virion?

Response: Thank you for pointing this out. There is indeed a prohead protease that

cleaves off the N-terminal part that is therefore also missing in the density. We have adapted the text accordingly and supplied the reference to the biochemical study that shows this. (<https://pmc.ncbi.nlm.nih.gov/articles/PMC3517404/>)

6. Ln180-182: What is the percent sequence identity and RMSD between the additional proteins of 7-7-1 and Milano?

Response: Thanks for this question. We have included identity comparisons between 771+Milano and 777+OLIVR4 (80% in both cases as determined by ANI) and have adapted the main text accordingly.

7. Ln184-187: Can the authors provide a figure or panel showing the disulfide bonds between the linker and MCPs? Are there examples of such covalent links in other known phage capsids?

Response: We have made Supplementary figure 4 to show the vast extent of the disulfide linkages in both the hexamer and pentamer conformations, as well as their differences. These differences are now also discussed in the main text. Comparable features have also been described for phage Milano. We believe we are the first to report the differences between the arrangements in the pentamer and hexamer.

8. Ln193-200 and Figure 2: Is the Ig-like domain of gp122 resolved in the map? Its location in the coordinate model of Figure 2a is unclear and the domain is colored grey instead of rainbow in 2b. Can 2c be labelled to emphasize the location of the Ig-like domain?

Response: We agree with the reviewer that the Ig-type domain is not clearly represented in figure 2 and the main text. The quality of the map is not sufficient to do any meaningful model building. Therefore, we have used AlphaFold to predict the

remaining LP2 Ig-type domain. This part is not included in the model in figure 2a as we cannot confidently place the structure in it. We have adapted the figure and main text to clarify this. Also, we have highlighted the fiber density and the Ig-type domain green and encircled the part of LP2 that is missing.

9. Are the authors able to reveal any additional structural information of the capsid-fiber base from either C1 reconstructions of the capsids or from symmetry-expansion methods such as those outlined here: <https://guide.cryosparc.com/processing-data/tutorials-and-case-studies/case-study-end-to-end-processing-of-encapsulated-ferritin-empiar-10716?>

The capsid fibers are a major focus of this manuscript and any additional structural detail(s) that could lead to the components identification would be impactful.

Response: It would indeed be of great impact if the fibers could be resolved. We have tried a whole range of reconstruction strategies including different symmetries C1, C5, I, and local refinements, symmetry relaxation as the reviewer described, 3D classifications, and 3D flexible refinements. Unfortunately, none of the above-mentioned strategies led to better density as shown in the icosahedral reconstruction. This indicates that the fibers are not only breaking with icosahedral symmetry but are also most likely of such a flexible nature that none of these strategies can lock onto it.

10. The T=9 capsid asymmetric unit has 8 hexameric MCP protomers and 1 pentameric MCP protomer, but the presented model consists of 12 hexameric MCP and 5 pentameric MCP protomers. Can the authors provide a rationale for modelling the components as presented?

Response: Thanks for the reviewer's question. The capsid model provided here does

indeed not portray the asymmetric unit that can be multiplied into the full capsid via matrices. We have chosen this representation as we wanted to portray a center hexamer, a connected side hexamer, and a connected pentamer, which can represent all the situations in a capsid, without the need for matrix operations. Especially, since the focus of this work is heavily on what is happening with the capsid fibers protruding from the pentamer and to a lesser extent to the full capsid, we reasoned that the pentamer needed to be strongly represented in the model.

11. Have the coordinate models for the capsid and tail been refined with the same symmetry relation among chains as enforced on the reconstruction?

Response: The full capsid map has been solved with icosahedral symmetry. The model does not represent the full capsid and therefore we have not applied symmetry to it. For the tail we did not specify a symmetry in model refinement.

12. Can the authors please add the map-vs-model FSC for the modelled components to SI figure 5, as well as add the FSC0.5 and refined ADPs values to the SI table?

Response: We have added the map-vs-model FCS curves, the FSC0.5 and the refined ADP values.

Suggestions for clarity:

Ln26-27: remove “,and built atomic models of capsid hexamers, pentamers and tail” since it reads redundantly to “determined capsid and tail structure”

- Ln55-57: I recommend omitting this sentence since it adds specifics on Agrobacter that are not further discussed in the manuscript.

- Figure 1:

- o a: Please indicate the length of the scale bar on the figure or in the legend.

- o The Caspar-Klug system (Caspar and Klug, 1962) might be a more appropriate reference than viralzone.expasy.org for triangulation number.

- Ln136-140: Milano is referenced twice as a close relative. Omit one of the references

- Ln349: remove '=' after "nature"

- Ln468: replace "xx" with value.

Response: We have implemented all the suggestions listed above.

Reviewer 4:

In the paper titled, Using cryogenic electron microscopy methods to gain insight into structure and initial host attachment of the flagellotropic bacteriophage 7-7-1, by Noteborn et al., the authors used cryo-electron microscopy, cryo-electron tomography and machine learning to examine phage-host interactions. They were able to visualize capsid fiber binding around the bacterial flagellum at higher resolutions than have previously been described. The authors also generated atomic models of the phage capsid and tail.

We thank the reviewer for the comments on our manuscript and for the constructive criticism provided, which we address in detail below.

Comments:

1. There are no page or line numbers, but the authors state around page 12, Intriguingly,

we found that the capsid fibers of phage 7-7-1 not only seem to interact with the host's flagellum but could also interact with fibers from other 7-7-1 capsids (Figure 4c, white arrow). I could not find anywhere that it was stated how many times the authors made this observation? I am wondering if this could also have happened by chance because the phage heads were near each other rather than showing an actual interaction. Is there any evidence that these capsid fibers interact with each other in other systems? Could the proteins that are predicted to exist on the termini be overexpressed and then examined directly? The reason I wonder if this is a real observation or an artifact is because these fibers would need to be expressed within the bacterial cytoplasm and if they bind to each other, it is difficult to imagine how they assemble with the correct stoichiometry at each annex and do not agglutinate.

Response: We thank the reviewer for this question. We found these interactions on multiple occasions. At this moment, we cannot rule out if this is due to an artifact of the cryo-EM technique or if this is real. We state in the main text that we are unsure what the type of interaction is that we are seeing, but as we haven't seen anything like this in other literature we believe it is of relevance to mention. We agree that it would be unlikely that there is a strong "designed" interaction as it would indeed cause the multiple fibers both on individual and neighboring capsids to clump together. We do believe that this interaction could be of a more transient nature like a physical entanglement. Unfortunately, we do not know the composition of the fibers and do not know the exact proteins involved. It would indeed be interesting for future work to define the nature of these interactions if the composition becomes known.

2. For Figure 6, I would argue that it is difficult to conclude that this assay is indicating the impact of the capsid fiber on bacterial infectivity. I do not think the authors have

any evidence that the capsid fibers are more sensitive to PK digestion than the tail fibers that are also important for host binding. Also, most of the phages shown after 60 min incubation with PK appear to look just like the empty capsid phages observed in Suppl Fig 1b and so these phages would also be unable to infect their host cells.

Response: We thank the reviewer for bringing up these arguments. We agree and have decided to remove the infectivity panel from the manuscript as it was not conclusive enough and did not highly contribute to the main text.

3. In the Materials and Methods section in the first paragraph, a number is missing in the following sentence: centrifuged 2-3 times at $5,000 \times g$ for 15 min each to approximately xx μl until no buffer eluted from the protein concentrator.

Response: We thank the reviewer for pointing this out. We have added the correct value.

Reviewer 5:

The authors present structural studies of bacteriophage 7-7-1, a phage that initiates infection by attaching to host flagella. Cryo-EM is employed to reconstruct the phage's capsid and tail structures at high resolution, drawing insight into the arrangements of pentameric and hexameric subunits, and enabling atomic modeling of their constituent proteins. To investigate capsid fibers, which are hypothesized to play a critical role in the initiation of infection, cryo-ET is additionally employed to visualize phages in the context of their flagellar interactions. Based on machine learning-guided segmentation of phage structures within tomograms, the authors posit that capsid fibers form the initial attachment between the phage and a flagellum, and may additionally be involved in the recruitment of other phage particles.

The manuscript provides structural insights into the plant bacteria-infecting bacteriophage 7-7-1, adding to, and providing additional validation for, structural studies conducted on other flagellotropic phages. The use of cryo-ET in conjunction with machine learning-based segmentation is a promising approach suggesting that capsid fibers play a critical role in phage attachment and recruitment. However, the employed machine learning architecture and training strategy, which the authors had developed in a previous publication, appear to produce capsid fiber segmentations that are suboptimal in quality. Improved prediction or visualization of these segmentations may clarify the evidence the authors present in favor of the functional hypotheses they propose.

Comments:

1. Compared to the manual capsid fiber segmentation presented in Figure 4, the automated segmentations presented in Figures 5 and S4 appear to be of noticeably lower quality. The fragmented nature of these automated segmentations make their visual interpretation challenging, especially in relation to observations of “capsid fibers extending from the phage and enveloping the flagellum.” Could the quality of the visualizations be improved by noise removal or manual cleanup? Alternatively, by performing cross-validation on your set of manual annotations, could you employ common evaluation metrics for cryo-ET segmentation such as the DICE and Surface-DICE scores discussed in Lamm et al. [1], in order to assess the accuracy of your trained model?

[1] Lamm et al. MemBrain v2: an end-to-end tool for the analysis of membranes in cryo-electron tomography. biorXiv 2024.

Response: We appreciate the reviewer’s feedback regarding the quality of the

automated segmentation results. The observed fragmentation in Figures 5 and S4 primarily arises from the inherent challenges of segmenting thin, low-contrast structures such as capsid fibers in cryo-electron tomograms. While manual segmentation (as shown in Figure 4) benefits from expert-driven experience, our automated approach is constrained by the variability in fiber visibility and tomographic noise. We have used IsoNet to enhance the clarity of the reconstructed tomograms before segmentation. Although of lesser quality compared to manual segmentation, the obtained results were still good enough to determine the average length of the fibers and their interaction with the flagella in a much less time-consuming manner.

2. An interesting hypothesis is presented that “an initial phage seeks out and attaches to a flagellum, and subsequently, another phage can locate and connect to the first phage via the capsid fibers, thus facilitating its proximity to the bacterial cell and subsequent completion of infection”. Since you have comprehensively segmented tomograms, you might consider highlighting these connections in a figure to provide further evidence.

We appreciate the reviewer’s suggestion. The hypothesis that an initial phage attaches to a flagellum and subsequently facilitates the attachment of additional phages via capsid fibers is indeed a fascinating aspect of our findings. While our segmentation approach allows for detailed tomographic reconstruction, directly visualizing these sequential interactions remains challenging due to resolution constraints and the dynamic nature of phage movement.

Nevertheless, we recognize the value of highlighting these connections to provide additional supporting evidence. To address this, we will include zoomed-in figures to better illustrate the connections between capsid fibers (SI Fig 6).

3. Is there a reason that the phage tails are omitted from the training of the automated

segmentation model? The annotation of tails in Figure 4 is helpful in visualizing phage orientation, and could be similarly helpful in Figure 5.

Response: Thank you for your suggestion. We agree that the phage tail is helpful in visualizing phage orientation. In Figure 4, the segmentation was performed manually; however, our neural network was trained specifically to segment phage fibers. Segmenting the tail is less effective because, unlike the fibers, the tail appears as a cylindrical structure with a larger size and greater diameter, making it challenging for this model to distinguish reliably.

The primary purpose of Figure 5 is to illustrate the diversity of capsid fibers and provide evidence that phage 771 can use its capsid to "climb" along flagella. Given this focus, we believe that segmenting the phage tail is not essential for the analysis presented in this figure.

List of Responses

Dear Editors and Reviewers:

Thank you for your constructive comments concerning our manuscript entitled " Using cryogenic electron microscopy methods to gain insight into structure and initial host attachment of the flagellotropic bacteriophage 7-7-1". We have addressed each input as outlined below, which has improved the manuscript significantly. The revised text is marked red in the manuscript to highlight the changes we've made. The main corrections in the manuscript and the responses to the reviewers' comments are outlined below.

Responds to the reviewers' comments:

Reviewer 1:

This is a well-written manuscript that advances the field of phage research by providing novel insights into flagellotropic phages, which have not been studied extensively. In particular, the authors use an innovative combination of cryo-EM SPA and ET methods in combination with machine learning, enabling them to track and segment the novel flexible capsid fibers, and directly visualizing the interaction between capsid and flagella, and also between capsids. Furthermore, they clearly demonstrate the time-dependent fiber degradation with a proteinase-K assay, which correlates well with infectivity estimated in a spot assay. The phage itself is a new member among only few examples of jumbo phages, which adds to the significance of this work.

The structure and role of viral capsid fibers is still not fully understood, and more research is necessary. This work pushes the frontier to the nature of flagellotropic

interactions. The tomography is excellent, as expected from this group. Although the resolution of the individual tomograms is limited due to the missing wedge, and subtomogram averaging is challenging due to the flexibility of the fibers, the gained insights describe a new paradigm how phages interact with flagella. I find the author's hypothesis about a collaborative phage network in proximity to flagella refreshing and stimulating. The paper will therefore appeal to a broad audience and is well suited for publication in Comms Bio.

We thank the reviewer for the kind remarks about our manuscript and for the useful comments, which we address in detail below.

Minor comments:

1. Fig.1 legend and also main text – “small threshold level of 0.01 in chimerax”: this value is not very informative, unless the reader is using the same software. The authors could normalize the map (average 0, stand dev 1) and specify the contour level as multiple of standard deviations.

Response:

We agree with the reviewers' comment. We have adapted the wording in the text for explaining the difference in contour levels (lines 122 - 125). In line with other reviewers' comments to make this display the capsid fibers better, we have adapted figure 2C, where we have better represented the capsid fiber, and have made SI figure 4 where we show enlarged side and top views of the fiber density.

2. If the genes for the fibers are known, they could be compared by sequence alignment with others such as in ϕ Cb13 and ϕ CbK. Could AlphaFold3 produce a meaningful structure?

Response: Unfortunately, the 7-7-1 genes are unknown to us and we could not find any capsid fiber sequence similarity between other species (including ϕ Cb13 and ϕ CbK). In a brute force attempt, we even predicted the entire proteome with AF3 and checked for candidate structures and interactions, but none were identified.

3. The tomograms should also be deposited to the EMDB.

Response: Two representative cryo-tomograms have been deposited to the EMDB under number EMD-53352. The main text has been modified to include this reference (lines 725 - 726).

Reviewer 2:

In this manuscript entitled “Using cryogenic electron microscopy methods to gain insight into the structure and initial host attachment of the flagellotropic bacteriophage 7-7-1”, Noteborn et al. determined the atomic structures of capsid and tail from bacteriophage 7-7-1 by cryo-EM. They also investigated the interactions between 7-7-1 and host flagellum by cryo-ET and Proteinase K treatment analysis.

All the results give us an insight into the architecture of bacteriophage 7-7-1 and describe the interactions involving capsid fiber and tail fiber. This study is solid due to its experimental design and mythology in the cryo-EM field. The experimental design is generally coherent, the methods are proper, and the results are meaningful. The more impressive part is that they use machine learning to pick up the phages and show different types of interactions.

We thank the reviewer for the positive comments on our manuscript and for the constructive criticism provided, which we address in detail below.

Main suggestions:

1. Page 12, Line 284-285: “Our SPA result has revealed that the fibers emerge from the 11 pentamers on the capsid. However...” The reconstruction of capsid relies on I symmetry (Supplementary Table 1), which means the densities of 12 pentamers have been averaged, including the “capsid fiber” (root part on Capsid). Thus, although it's highly possible that each pentamer vertex has a capsid fiber (except the tail vertex), the SPA structure alone can not prove that point. Unless there is other evidence.

Response: Thank you for pointing this out- this indeed needs to be further elaborated. Indeed, SPA alone would not prove the presence of a capsid fiber at each pentamer vertex given the I symmetry. However, the combination of SPA and tomography supports this conclusion: In the tomograms we can clearly see fibers emerging from all pentamers (unless the density is obscured by the ‘missing wedge’, the imaging artifact that originates due to the incomplete tilt range during data collection). We have adapted the text accordingly (lines 447-451).

2. In Figure 1a and Figure 3a, adding a length marker could give the reader a direct feeling about the size.

Response: Thanks for the suggestion. We have added scale bars with numbers to all relevant figures.

3. In Figure 6a, “no treatment,” “20 min,” “40 min,” and “60 min” could be added in the left corner of each image, which will be much clearer.

Response: Thanks for the suggestion. We have adapted the figure accordingly.

4. In Figure 6b, why the “20 min” treatment data is missing?

Response: After internal consideration we have decided to remove the panel on infectivity from the manuscript as it was not conclusive enough and did not highly contribute to the main text. We do thank the reviewer for their comment.

5. Page 19, Line 468, “xx ul,” xx should be a number.

Response: Thanks for pointing this out. We have added the correct number of microliters (75 ul) (line 605).

6. Although it’s a supplementary figure, Supplementary Figure 1 still needs scale bars in the figure.

Response: Thank you for pointing this out. We have added scale bars to this figure.

Reviewer 3:

Noteborn et al. use cryo-electron microscopy, machine-learning, and molecular biology techniques to investigate the structural basis of host cell attachment by bacteriophage 7-7-1. The approach is similar to the authors’ previous work (Ouyang et al., 2022), but is nonetheless distinct as here the methods are used to study capsid fibers rather than tail fibers. The results here will certainly be of interest to readers studying viruses using similar mechanisms of host-attachment, as well as to structural virologists for the methods employed by the authors.

Nevertheless, I have questions and suggestions for the authors to consider that I hope will add clarity to the manuscript prior to its publication.

We thank the reviewer for the kind remarks about our manuscript and for the useful comments, which we address in detail below.

Comments:

1. Ln70-71: Are the authors able to add specifics to the statements in these lines?

What is the percent identity between 7-7-1 and Milano/OLIVR4 bacteriophages?

What is considered an “unusual accumulation of cysteine residues”? What is the percentage of cysteine residues in 7-7-1 structural proteins relative to other Myoviridae of similar genome size?

Response: We thank the reviewer for this comment and agree more elaboration is necessary. We have added the specifics on the percent identity between the different species (80% in both cases) and have gone deeper into the importance of the cysteine residue accumulation (1.45% whole genome, 2.8% in the structural proteins studied here) (lines 65-70). This will also follow in answer to comment #7.

2. Ln119-122: Can the authors prepare a main-text figure panel and/or supplementary figure showing an enlarged view of the reconstructed capsid fiber density? The capsid fibers are a major point in the manuscript and such a figure would help readers appreciate how difficult they are to interpret from the icosahedral averaged reconstruction.

Response: Thank you for this comment. We agree with the reviewer that the capsid fiber density was too poorly represented. We have adapted figure 2 and made a new supplementary figure 4 to better represent the capsid fiber density.

3. Ln136-140: In addition to the same T-number, do all the phages share a similar

HK97-fold topology and capsid size?

Response: Thanks for the reviewer's question. According to the referenced studies, in addition to sharing the same $T = 9$ number, the major capsid proteins of all the listed phages (*Ralstonia solanacearum* phage GP4, satellite phage P2, phage N4, *Anabaena* phage A-1(L), *Helicobacter pylori* phages KHP30 and KHP40, and phage Milano) exhibit a similar HK97-fold topology. However, they do not share identical capsid sizes. For example, the diameter of phage P2 is approximately 60 nm, phage GP4 is 74 nm, phage KHP30 is 70 nm, and our phage 7-7-1 is about 80 nm. We have adapted the main text to reflect these statements (lines 132-134).

4. Ln162-164: What is the percent identity and RMSD between the 7-7-1 MCP and the canonical HK97-fold? Can the authors also provide a main text figure panel or SI figure with the 7-7-1 MCP labelled according to the conventional HK97-fold nomenclature (i.e., A-domain, P-domain, E-loop, etc.)? This can be added SI Figure 2, as it will help to highlight the major difference of hexamer and pentamer protomers conformations in the E-loop.

Response: We thank the reviewer for this comment that will indeed clarify the structural features of the MCP. We have added the HK97 fold labelling and the comparison with the classical HK97 fold and the RMSD between the two structures to Supplementary Figure 2. We have also discussed the differences in the N-arm and E-loop between the hexamer and pentamer conformation in the main text (lines 145-149, 154-155).

5. Ln166-168: Are the residues unresolved because of high flexibility or is the MCP expressed in a pro-form that is matured through cleavage by a prohead protease like in other phages? Does this phage encode a prohead protease? Is there proteomics or SDS-PAGE of virions to suggest proteolysis of the MCP in the mature virion?

Response: Thank you for pointing this out. There is indeed a prohead protease that cleaves off the N-terminal part that is therefore also missing in the density. We have adapted the text accordingly (lines 147-149) and supplied the reference to the biochemical study that shows this.

[\(https://pmc.ncbi.nlm.nih.gov/articles/PMC3517404/\)](https://pmc.ncbi.nlm.nih.gov/articles/PMC3517404/).

6. Ln180-182: What is the percent sequence identity and RMSD between the additional proteins of 7-7-1 and Milano?

Response: Thanks for this question. We have included identity comparisons between 771+Milano and 777+OLIVR4 (80% in both cases as determined by ANI) and have adapted the main text accordingly (lines 65-70). Additionally, we have written an entire section where we compare the RMSD and TM-score of all proteins identified here with Milano, E217 and T4 (lines 383-405).

7. Ln184-187: Can the authors provide a figure or panel showing the disulfide bonds between the linker and MCPs? Are there examples of such covalent links in other known phage capsids?

Response: We have made Supplementary figure 3 to show the vast extent of the disulfide linkages in both the hexamer and pentamer confirmations, as well as their differences. These differences are now also discussed in the main text (lines 194-209). Comparable features have also been described for phage Milano. We believe we are the first to report the differences between the arrangements in the pentamer and hexamer.

8. Ln193-200 and Figure 2: Is the Ig-like domain of gp122 resolved in the map? Its location in the coordinate model of Figure 2a is unclear and the domain is colored grey instead of rainbow in 2b. Can 2c be labelled to emphasize the location of the Ig-like

domain?

Response: We agree with the reviewer that the Ig-type domain is not clearly represented in figure 2 and the main text. The quality of the map is not sufficient to do any meaningful model building. Therefore, we have used AlphaFold to predict the remaining LP2 Ig-type domain. This part is not included in the model in figure 2a as we cannot confidently place the structure in it. We have adapted the figure and main text to clarify this (lines 182-188). Also, we have highlighted the fiber density and the Ig-type domain green and encircled the part of LP2 that is missing.

9. Are the authors able to reveal any additional structural information of the capsid-fiber base from either C1 reconstructions of the capsids or from symmetry-expansion methods such as those outlined here: <https://guide.cryosparc.com/processing-data/tutorials-and-case-studies/case-study-end-to-end-processing-of-encapsulated-ferritin-empiar-10716>?

The capsid fibers are a major focus of this manuscript and any additional structural detail(s) that could lead to the components identification would be impactful.

Response: It would indeed be of great impact if the fibers could be resolved. We have tried a whole range of reconstruction strategies including different symmetries C1, C5, I, and local refinements, symmetry relaxation as the reviewer described, 3D classifications, and 3D flexible refinements. Unfortunately, none of the above-mentioned strategies led to better density as shown in the icosahedral reconstruction. This indicates that the fibers are not only breaking with icosahedral symmetry but are also most likely of such a flexible nature that none of these strategies can lock onto it.

10. I highly recommend the authors deposit the data on EMPIAR for general data transparency, as well as encouraging methods development on difficult targets such as the capsid fibers presented in the manuscript.

Response: The tomography data has been submitted to EMPIAR under accession code EMPIAR-13014.

11. The T=9 capsid asymmetric unit has 8 hexameric MCP protomers and 1 pentameric MCP protomer, but the presented model consists of 12 hexameric MCP and 5 pentameric MCP protomers. Can the authors provide a rationale for modelling the components as presented?

Response: Thanks for the reviewer's question. The capsid model provided here does indeed not portray the asymmetric unit that can be multiplied into the full capsid via matrices. We have chosen this representation as we wanted to portray a center hexamer, a connected side hexamer, and a connected pentamer, which can represent all the situations in a capsid, without the need for matrix operations. Especially, since the focus of this work is heavily on what is happening with the capsid fibers protruding from the pentamer and to a lesser extent to the full capsid, we reasoned that the pentamer needed to be strongly represented in the model.

12. Have the coordinate models for the capsid and tail been refined with the same symmetry relation among chains as enforced on the reconstruction?

Response: The full capsid map has been solved with icosahedral symmetry. The model does not represent the full capsid and therefore we have not applied symmetry to it. For the tail we did not specify a symmetry in model refinement.

13. Can the authors please add the map-vs-model FSC for the modelled components to

SI figure 5, as well as add the FSC0.5 and refined ADPs values to the SI table?

Response: We have added the map-vs-model FCS curves (Supplementary figure 10), the FSC0.5 (Supplementary table 1) and the refined ADP values (Supplementary table 3).

Suggestions for clarity:

Ln26-27: remove “,and built atomic models of capsid hexamers, pentamers and tail” since it reads redundantly to “determined capsid and tail structure”

- Ln55-57: I recommend omitting this sentence since it adds specifics on Agrobacter that are not further discussed in the manuscript.

- Figure 1:

- o a: Please indicate the length of the scale bar on the figure or in the legend.

- o The Caspar-Klug system (Caspar and Klug, 1962) might be a more appropriate reference than viralzone.expasy.org for triangulation number.

- Ln136-140: Milano is referenced twice as a close relative. Omit one of the references

- Ln349: remove ‘=’ after “nature”

- Ln468: replace “xx” with value.

Response: We have implemented all the suggestions listed above.

Reviewer 4:

In the paper titled, Using cryogenic electron microscopy methods to gain insight into structure and initial host attachment of the flagellotropic bacteriophage 7-7-1, by Noteborn et al., the authors used cryo-electron microscopy, cryo-electron tomography and machine learning to examine phage-host interactions. They were able to visualize capsid fiber binding around the bacterial flagellum at higher resolutions than have previously been described. The authors also generated atomic models of the phage capsid and tail.

We thank the reviewer for the comments on our manuscript and for the constructive criticism provided, which we address in detail below.

Comments:

1. There are no page or line numbers, but the authors state around page 12, Intriguingly, we found that the capsid fibers of phage 7-7-1 not only seem to interact with the host's flagellum but could also interact with fibers from other 7-7-1 capsids (Figure 4c, white arrow). I could not find anywhere that it was stated how many times the authors made this observation? I am wondering if this could also have happened by chance because the phage heads were near each other rather than showing an actual interaction. Is there any evidence that these capsid fibers interact with each other in other systems? Could the proteins that are predicted to exist on the termini be overexpressed and then examined directly? The reason I wonder if this is a real observation or an artifact is because these fibers would need to be expressed within the bacterial cytoplasm and if they bind to each other, it is difficult to imagine how they assemble with the correct stoichiometry at each annex and do not agglutinate.

Response: We thank the reviewer for this question. We found these interactions on multiple occasions (lines 410-411, 450-451). At this moment, we cannot rule out if this is due to an artifact of the cryo-EM technique or if this is real. We state in the main text that we are unsure what the type of interaction is that we are seeing, but as we haven't seen anything like this in other literature, we believe it is of relevance to mention. We agree that it would be unlikely that there is a strong "designed" interaction as it would indeed cause the multiple fibers both on individual and neighboring capsids to clump together. We do believe that this interaction could be of a more transient nature like a physical entanglement. Unfortunately, we do not know the composition of the fibers and do not know the exact proteins involved. It would indeed be interesting for future work to define the nature of these interactions if the composition becomes known.

2. For Figure 6, I would argue that it is difficult to conclude that this assay is indicating the impact of the capsid fiber on bacterial infectivity. I do not think the authors have any evidence that the capsid fibers are more sensitive to PK digestion than the tail fibers that are also important for host binding. Also, most of the phages shown after 60 min incubation with PK appear to look just like the empty capsid phages observed in Suppl Fig 1b and so these phages would also be unable to infect their host cells.

Response: We thank the reviewer for bringing up these arguments. We agree and have decided to remove the infectivity panel from the manuscript as it was not conclusive enough and did not highly contribute to the main text.

3. In the Materials and Methods section in the first paragraph, a number is missing in the following sentence: centrifuged 2-3 times at $5,000 \times g$ for 15 min each to approximately xx μ l until no buffer eluted from the protein concentrator.

Response: We have added the correct value (line 605).

Reviewer 5:

The authors present structural studies of bacteriophage 7-7-1, a phage that initiates infection by attaching to host flagella. Cryo-EM is employed to reconstruct the phage's capsid and tail structures at high resolution, drawing insight into the arrangements of pentameric and hexameric subunits, and enabling atomic modeling of their constituent proteins. To investigate capsid fibers, which are hypothesized to play a critical role in the initiation of infection, cryo-ET is additionally employed to visualize phages in the context of their flagellar interactions. Based on machine learning-guided segmentation of phage structures within tomograms, the authors posit that capsid fibers form the initial attachment between the phage and a flagellum, and may additionally be involved in the recruitment of other phage particles.

The manuscript provides structural insights into the plant bacteria-infecting bacteriophage 7-7-1, adding to, and providing additional validation for, structural studies conducted on other flagellotropic phages. The use of cryo-ET in conjunction with machine learning-based segmentation is a promising approach suggesting that capsid fibers play a critical role in phage attachment and recruitment. However, the employed machine learning architecture and training strategy, which the authors had developed in a previous publication, appear to produce capsid fiber segmentations that are suboptimal in quality. Improved prediction or visualization of these segmentations may clarify the evidence the authors present in favor of the functional hypotheses they propose.

Comments:

1. Compared to the manual capsid fiber segmentation presented in Figure 4, the

automated segmentations presented in Figures 5 and S4 appear to be of noticeably lower quality. The fragmented nature of these automated segmentations make their visual interpretation challenging, especially in relation to observations of “capsid fibers extending from the phage and enveloping the flagellum.” Could the quality of the visualizations be improved by noise removal or manual cleanup? Alternatively, by performing cross-validation on your set of manual annotations, could you employ common evaluation metrics for cryo-ET segmentation such as the DICE and Surface-DICE scores discussed in Lamm et al. [1], in order to assess the accuracy of your trained model?

[1] Lamm et al. MemBrain v2: an end-to-end tool for the analysis of membranes in cryo-electron tomography. *bioRxiv* 2024.

Response: We appreciate the reviewer’s feedback regarding the quality of the automated segmentation results. The observed fragmentation in Figures 5 and S4 (now Figure 7 and S8) primarily arises from the inherent challenges of segmenting thin, low-contrast structures such as capsid fibers in cryo-electron tomograms. While manual segmentation (as shown in Figure 6) benefits from expert-driven experience, our automated approach is constrained by the variability in fiber visibility and tomographic noise. We have used IsoNet to enhance the clarity of the reconstructed tomograms before segmentation. Although of lesser quality compared to manual segmentation, the obtained results were still good enough to determine the average length of the fibers and their interaction with the flagella in a much less time-consuming manner.

2. An interesting hypothesis is presented that “an initial phage seeks out and attaches to a flagellum, and subsequently, another phage can locate and connect to the first phage via the capsid fibers, thus facilitating its proximity to the bacterial cell and subsequent

completion of infection". Since you have comprehensively segmented tomograms, you might consider highlighting these connections in a figure to provide further evidence.

We appreciate the reviewer's suggestion. The hypothesis that an initial phage attaches to a flagellum and subsequently facilitates the attachment of additional phages via capsid fibers is indeed a fascinating aspect of our findings. While our segmentation approach allows for detailed tomographic reconstruction, directly visualizing these sequential interactions remains challenging due to resolution constraints and the dynamic nature of phage movement.

Nevertheless, we recognize the value of highlighting these connections to provide additional supporting evidence. To address this, we will include zoomed-in figures to better illustrate the connections between capsid fibers (SI Fig 6).

3. Is there a reason that the phage tails are omitted from the training of the automated segmentation model? The annotation of tails in Figure 4 is helpful in visualizing phage orientation, and could be similarly helpful in Figure 5.

Response: Thank you for your suggestion. We agree that the phage tail is helpful in visualizing phage orientation. In Figure 6 (previously Fig4), the segmentation was performed manually; however, our neural network was trained specifically to segment phage fibers. Segmenting the tail is less effective because, unlike the fibers, the tail appears as a cylindrical structure with a larger size and greater diameter, making it challenging for this model to distinguish reliably.

The primary purpose of Figure 7 (previously Fig5) is to illustrate the diversity of capsid fibers and provide evidence that phage 771 can use its capsid to "climb" along flagella. Given this focus, we believe that segmenting the phage tail is not essential for the

analysis presented in this figure.

List of Responses

Dear Editors and Reviewers:

Thank you for your constructive comments concerning our manuscript entitled " Using cryogenic electron microscopy methods to gain insight into structure and initial host attachment of the flagellotropic bacteriophage 7-7-1". We have addressed each input as outlined below, which has improved the manuscript significantly. The revised text is marked red in the manuscript to highlight the changes we've made. The main corrections in the manuscript and the responses to the reviewers' comments are outlined below.

Data availability:

The EMDB/PDB structures, maps and representative tomogram (and the corresponding validation reports) are online in a dedicated researchdrive folder and are directly accessible via the link below:

<https://universiteitleiden.data.surf.nl/s/7CycMi9LfBgYcYz>

The raw data is accessible via the EMPIAR reviewers invitation option:

username: review_d6343f98

password: 91ffbe9b

Responds to the reviewers' comments:

Reviewer 1:

This is a well-written manuscript that advances the field of phage research by providing novel insights into flagellotropic phages, which have not been studied extensively. In particular, the authors use an innovative combination of cryo-EM SPA and ET methods in combination with machine learning, enabling them to track and segment the novel flexible capsid fibers, and directly visualizing the interaction between capsid and flagella, and also between capsids. Furthermore, they clearly demonstrate the time-dependent fiber degradation with a proteinase-K assay, which correlates well with infectivity estimated in a spot assay. The phage itself is a new member among only few examples of jumbo phages, which adds to the significance of this work.

The structure and role of viral capsid fibers is still not fully understood, and more research is necessary. This work pushes the frontier to the nature of flagellotropic interactions. The tomography is excellent, as expected from this group. Although the resolution of the individual tomograms is limited due to the missing wedge, and subtomogram averaging is challenging due to the flexibility of the fibers, the gained insights describe a new paradigm how phages interact with flagella. I find the author's hypothesis about a collaborative phage network in proximity to flagella refreshing and stimulating. The paper will therefore appeal to a broad audience and is well suited for publication in Comms Bio.

We thank the reviewer for the kind remarks about our manuscript and for the useful comments, which we address in detail below.

Minor comments:

1. Fig.1 legend and also main text – “small threshold level of 0.01 in chimerax”: this value is not very informative, unless the reader is using the same software. The authors could normalize the map (average 0, stand dev 1) and specify the contour level as

multiple of standard deviations.

Response:

We agree with the reviewers' comment. We have adapted the wording in the text for explaining the difference in contour levels (lines 122 - 125). In line with other reviewers' comments to make this display the capsid fibers better, we have adapted figure 2C, where we have better represented the capsid fiber, and have made SI figure 4 where we show enlarged side and top views of the fiber density.

2. If the genes for the fibers are known, they could be compared by sequence alignment with others such as in ϕ Cb13 and ϕ CbK. Could AlphaFold3 produce a meaningful structure?

Response: Unfortunately, the 7-7-1 genes are unknown to us and we could not find any capsid fiber sequence similarity between other species (including ϕ Cb13 and ϕ CbK). In a brute force attempt, we even predicted the entire proteome with AF3 and checked for candidate structures and interactions, but none were identified.

3. The tomograms should also be deposited to the EMDB.

Response: Two representative cryo-tomograms have been deposited to the EMDB under number EMD-53352. The main text has been modified to include this reference (lines 725 - 726).

Reviewer 2:

In this manuscript entitled "Using cryogenic electron microscopy methods to gain insight into the structure and initial host attachment of the flagellotropic bacteriophage 7-7-1", Noteborn et al. determined the atomic structures of capsid and tail from

bacteriophage 7-7-1 by cryo-EM. They also investigated the interactions between 7-7-1 and host flagellum by cryo-ET and Proteinase K treatment analysis.

All the results give us an insight into the architecture of bacteriophage 7-7-1 and describe the interactions involving capsid fiber and tail fiber. This study is solid due to its experimental design and methodology in the cryo-EM field. The experimental design is generally coherent, the methods are proper, and the results are meaningful. The more impressive part is that they use machine learning to pick up the phages and show different types of interactions.

We thank the reviewer for the positive comments on our manuscript and for the constructive criticism provided, which we address in detail below.

Main suggestions:

1. Page 12, Line 284-285: “Our SPA result has revealed that the fibers emerge from the 11 pentamers on the capsid. However...” The reconstruction of capsid relies on I symmetry (Supplementary Table 1), which means the densities of 12 pentamers have been averaged, including the “capsid fiber” (root part on Capsid). Thus, although it's highly possible that each pentamer vertex has a capsid fiber (except the tail vertex), the SPA structure alone can not prove that point. Unless there is other evidence.

Response: Thank you for pointing this out- this indeed needs to be further elaborated. Indeed, SPA alone would not prove the presence of a capsid fiber at each pentamer vertex given the I symmetry. However, the combination of SPA and tomography supports this conclusion: In the tomograms we can clearly see fibers emerging from all

pentamers (unless the density is obscured by the ‘missing wedge’, the imaging artifact that originates due to the incomplete tilt range during data collection). We have adapted the text accordingly (lines 447-451).

2. In Figure 1a and Figure 3a, adding a length marker could give the reader a direct feeling about the size.

Response: Thanks for the suggestion. We have added scale bars with numbers to all relevant figures.

3. In Figure 6a, “no treatment,” “20 min”, “40 min,” and “60 min” could be added in the left corner of each image, which will be much clearer.

Response: Thanks for the suggestion. We have adapted the figure accordingly.

4. In Figure 6b, why the “20 min” treatment data is missing?

Response: After internal consideration we have decided to remove the panel on infectivity from the manuscript as it was not conclusive enough and did not highly contribute to the main text. We do thank the reviewer for their comment.

5. Page 19, Line 468, “xx ul,” xx should be a number.

Response: Thanks for pointing this out. We have added the correct number of microliters (75 ul) (line 605).

6. Although it’s a supplementary figure, Supplementary Figure 1 still needs scale bars in the figure.

Response: Thank you for pointing this out. We have added scale bars to this figure.

Reviewer 3:

Noteborn et al. use cryo-electron microscopy, machine-learning, and molecular biology techniques to investigate the structural basis of host cell attachment by bacteriophage 7-7-1. The approach is similar to the authors' previous work (Ouyang et al., 2022), but is nonetheless distinct as here the methods are used to study capsid fibers rather than tail fibers. The results here will certainly be of interest to readers studying viruses using similar mechanisms of host-attachment, as well as to structural virologists for the methods employed by the authors.

Nevertheless, I have questions and suggestions for the authors to consider that I hope will add clarity to the manuscript prior to its publication.

We thank the reviewer for the kind remarks about our manuscript and for the useful comments, which we address in detail below.

Comments:

1. Ln70-71: Are the authors able to add specifics to the statements in these lines?

What is the percent identity between 7-7-1 and Milano/OLIVR4 bacteriophages?

What is considered an "unusual accumulation of cysteine residues"? What is the percentage of cysteine residues in 7-7-1 structural proteins relative to other Myoviridae of similar genome size?

Response: We thank the reviewer for this comment and agree more elaboration is necessary. We have added the specifics on the percent identity between the different species (80% in both cases) and have gone deeper into the importance of the cysteine residue accumulation (1.45% whole genome, 2.8% in the structural proteins studied

here) (lines 65-70). This will also follow in answer to comment #7.

2. Ln119-122: Can the authors prepare a main-text figure panel and/or supplementary figure showing an enlarged view of the reconstructed capsid fiber density? The capsid fibers are a major point in the manuscript and such a figure would help readers appreciate how difficult they are to interpret from the icosahedral averaged reconstruction.

Response: Thank you for this comment. We agree with the reviewer that the capsid fiber density was too poorly represented. We have adapted figure 2 and made a new supplementary figure 4 to better represent the capsid fiber density.

3. Ln136-140: In addition to the same T-number, do all the phages share a similar HK97-fold topology and capsid size?

Response: Thanks for the reviewer's question. According to the referenced studies, in addition to sharing the same $T = 9$ number, the major capsid proteins of all the listed phages (*Ralstonia solanacearum* phage GP4, satellite phage P2, phage N4, *Anabaena* phage A-1(L), *Helicobacter pylori* phages KHP30 and KHP40, and phage Milano) exhibit a similar HK97-fold topology. However, they do not share identical capsid sizes. For example, the diameter of phage P2 is approximately 60 nm, phage GP4 is 74 nm, phage KHP30 is 70 nm, and our phage 7-7-1 is about 80 nm. We have adapted the main text to reflect these statements (lines 132-134).

4. Ln162-164: What is the percent identity and RMSD between the 7-7-1 MCP and the canonical HK97-fold? Can the authors also provide a main text figure panel or SI figure with the 7-7-1 MCP labelled according to the conventional HK97-fold nomenclature (i.e., A-domain, P-domain, E-loop, etc.)? This can be added SI Figure 2, as it will help

to highlight the major difference of hexamer and pentamer protomers conformations in the E-loop.

Response: We thank the reviewer for this comment that will indeed clarify the structural features of the MCP. We have added the HK97 fold labelling and the comparison with the classical HK97 fold and the RMSD between the two structures to Supplementary Figure 2. We have also discussed the differences in the N-arm and E-loop between the hexamer and pentamer conformation in the main text (lines 145-149, 154-155).

5. Ln166-168: Are the residues unresolved because of high flexibility or is the MCP expressed in a pro-form that is matured through cleavage by a prohead protease like in other phages? Does this phage encode a prohead protease? Is there proteomics or SDS-PAGE of virions to suggest proteolysis of the MCP in the mature virion?

Response: Thank you for pointing this out. There is indeed a prohead protease that cleaves off the N-terminal part that is therefore also missing in the density. We have adapted the text accordingly (lines 147-149) and supplied the reference to the biochemical study that shows this.

[\(https://pmc.ncbi.nlm.nih.gov/articles/PMC3517404/\)](https://pmc.ncbi.nlm.nih.gov/articles/PMC3517404/).

6. Ln180-182: What is the percent sequence identity and RMSD between the additional proteins of 7-7-1 and Milano?

Response: Thanks for this question. We have included identity comparisons between 771+Milano and 777+OLIVR4 (80% in both cases as determined by ANI) and have adapted the main text accordingly (lines 65-70). Additionally, we have written an entire section where we compare the RMSD and TM-score of all proteins identified here with Milano, E217 and T4 (lines 383-405).

7. Ln184-187: Can the authors provide a figure or panel showing the disulfide bonds between the linker and MCPs? Are there examples of such covalent links in other known phage capsids?

Response: We have made Supplementary figure 3 to show the vast extent of the disulfide linkages in both the hexamer and pentamer confirmations, as well as their differences. These differences are now also discussed in the main text (lines 194-209). Comparable features have also been described for phage Milano. We believe we are the first to report the differences between the arrangements in the pentamer and hexamer.

8. Ln193-200 and Figure 2: Is the Ig-like domain of gp122 resolved in the map? Its location in the coordinate model of Figure 2a is unclear and the domain is colored grey instead of rainbow in 2b. Can 2c be labelled to emphasize the location of the Ig-like domain?

Response: We agree with the reviewer that the Ig-type domain is not clearly represented in figure 2 and the main text. The quality of the map is not sufficient to do any meaningful model building. Therefore, we have used AlphaFold to predict the remaining LP2 Ig-type domain. This part is not included in the model in figure 2a as we cannot confidently place the structure in it. We have adapted the figure and main text to clarify this (lines 182-188). Also, we have highlighted the fiber density and the Ig-type domain green and encircled the part of LP2 that is missing.

9. Are the authors able to reveal any additional structural information of the capsid-fiber base from either C1 reconstructions of the capsids or from symmetry-expansion methods such as those outlined here: <https://guide.cryosparc.com/processing-data/tutorials-and-case-studies/case-study-end-to-end-processing-of-encapsulated-ferritin-empiar-10716>?

The capsid fibers are a major focus of this manuscript and any additional structural detail(s) that could lead to the components identification would be impactful.

Response: It would indeed be of great impact if the fibers could be resolved. We have tried a whole range of reconstruction strategies including different symmetries C1, C5, I, and local refinements, symmetry relaxation as the reviewer described, 3D classifications, and 3D flexible refinements. Unfortunately, none of the above-mentioned strategies led to better density as shown in the icosahedral reconstruction. This indicates that the fibers are not only breaking with icosahedral symmetry but are also most likely of such a flexible nature that none of these strategies can lock onto it.

10. I highly recommend the authors deposit the data on EMPIAR for general data transparency, as well as encouraging methods development on difficult targets such as the capsid fibers presented in the manuscript.

Response: The tomography data has been submitted to EMPIAR under accession code EMPIAR-13014.

11. The T=9 capsid asymmetric unit has 8 hexameric MCP protomers and 1 pentameric MCP protomer, but the presented model consists of 12 hexameric MCP and 5 pentameric MCP protomers. Can the authors provide a rationale for modelling the components as presented?

Response: Thanks for the reviewer's question. The capsid model provided here does indeed not portray the asymmetric unit that can be multiplied into the full capsid via matrices. We have chosen this representation as we wanted to portray a center hexamer, a connected side hexamer, and a connected pentamer, which can represent all the

situations in a capsid, without the need for matrix operations. Especially, since the focus of this work is heavily on what is happening with the capsid fibers protruding from the pentamer and to a lesser extent to the full capsid, we reasoned that the pentamer needed to be strongly represented in the model.

12. Have the coordinate models for the capsid and tail been refined with the same symmetry relation among chains as enforced on the reconstruction?

Response: The full capsid map has been solved with icosahedral symmetry. The model does not represent the full capsid and therefore we have not applied symmetry to it. For the tail we did not specify a symmetry in model refinement.

13. Can the authors please add the map-vs-model FSC for the modelled components to SI figure 5, as well as add the FSC0.5 and refined ADPs values to the SI table?

Response: We have added the map-vs-model FCS curves (Supplementary figure 10), the FSC0.5 (Supplementary table 1) and the refined ADP values (Supplementary table 3).

Suggestions for clarity:

Ln26-27: remove “,and built atomic models of capsid hexamers, pentamers and tail” since it reads redundantly to “determined capsid and tail structure”

- Ln55-57: I recommend omitting this sentence since it adds specifics on Agrobacter that are not further discussed in the manuscript.

- Figure 1:

- o a: Please indicate the length of the scale bar on the figure or in the legend.

o The Caspar-Klug system (Caspar and Klug, 1962) might be a more appropriate reference than viralzone.expasy.org for triangulation number.

• Ln136-140: Milano is referenced twice as a close relative. Omit one of the references

• Ln349: remove '=' after "nature"

• Ln468: replace "xx" with value.

Response: We have implemented all the suggestions listed above.

Reviewer 4:

In the paper titled, Using cryogenic electron microscopy methods to gain insight into structure and initial host attachment of the flagellotropic bacteriophage 7-7-1, by Noteborn et al., the authors used cryo-electron microscopy, cryo-electron tomography and machine learning to examine phage-host interactions. They were able to visualize capsid fiber binding around the bacterial flagellum at higher resolutions than have previously been described. The authors also generated atomic models of the phage capsid and tail.

We thank the reviewer for the comments on our manuscript and for the constructive criticism provided, which we address in detail below.

Comments:

1. There are no page or line numbers, but the authors state around page 12, Intriguingly,

we found that the capsid fibers of phage 7-7-1 not only seem to interact with the host's flagellum but could also interact with fibers from other 7-7-1 capsids (Figure 4c, white arrow). I could not find anywhere that it was stated how many times the authors made this observation? I am wondering if this could also have happened by chance because the phage heads were near each other rather than showing an actual interaction. Is there any evidence that these capsid fibers interact with each other in other systems? Could the proteins that are predicted to exist on the termini be overexpressed and then examined directly? The reason I wonder if this is a real observation or an artifact is because these fibers would need to be expressed within the bacterial cytoplasm and if they bind to each other, it is difficult to imagine how they assemble with the correct stoichiometry at each annex and do not agglutinate.

Response: We thank the reviewer for this question. We found these interactions on multiple occasions (lines 410-411, 450-451). At this moment, we cannot rule out if this is due to an artifact of the cryo-EM technique or if this is real. We state in the main text that we are unsure what the type of interaction is that we are seeing, but as we haven't seen anything like this in other literature, we believe it is of relevance to mention. We agree that it would be unlikely that there is a strong "designed" interaction as it would indeed cause the multiple fibers both on individual and neighboring capsids to clump together. We do believe that this interaction could be of a more transient nature like a physical entanglement. Unfortunately, we do not know the composition of the fibers and do not know the exact proteins involved. It would indeed be interesting for future work to define the nature of these interactions if the composition becomes known.

2. For Figure 6, I would argue that it is difficult to conclude that this assay is indicating the impact of the capsid fiber on bacterial infectivity. I do not think the authors have

any evidence that the capsid fibers are more sensitive to PK digestion than the tail fibers that are also important for host binding. Also, most of the phages shown after 60 min incubation with PK appear to look just like the empty capsid phages observed in Suppl Fig 1b and so these phages would also be unable to infect their host cells.

Response: We thank the reviewer for bringing up these arguments. We agree and have decided to remove the infectivity panel from the manuscript as it was not conclusive enough and did not highly contribute to the main text.

3. In the Materials and Methods section in the first paragraph, a number is missing in the following sentence: centrifuged 2-3 times at $5,000 \times g$ for 15 min each to approximately xx μ l until no buffer eluted from the protein concentrator.

Response: We have added the correct value (line 605).

Reviewer 5:

The authors present structural studies of bacteriophage 7-7-1, a phage that initiates infection by attaching to host flagella. Cryo-EM is employed to reconstruct the phage's capsid and tail structures at high resolution, drawing insight into the arrangements of pentameric and hexameric subunits, and enabling atomic modeling of their constituent proteins. To investigate capsid fibers, which are hypothesized to play a critical role in the initiation of infection, cryo-ET is additionally employed to visualize phages in the context of their flagellar interactions. Based on machine learning-guided segmentation of phage structures within tomograms, the authors posit that capsid fibers form the initial attachment between the phage and a flagellum, and may additionally be involved in the recruitment of other phage particles.

The manuscript provides structural insights into the plant bacteria-infecting

bacteriophage 7-7-1, adding to, and providing additional validation for, structural studies conducted on other flagellotropic phages. The use of cryo-ET in conjunction with machine learning-based segmentation is a promising approach suggesting that capsid fibers play a critical role in phage attachment and recruitment. However, the employed machine learning architecture and training strategy, which the authors had developed in a previous publication, appear to produce capsid fiber segmentations that are suboptimal in quality. Improved prediction or visualization of these segmentations may clarify the evidence the authors present in favor of the functional hypotheses they propose.

Comments:

1. Compared to the manual capsid fiber segmentation presented in Figure 4, the automated segmentations presented in Figures 5 and S4 appear to be of noticeably lower quality. The fragmented nature of these automated segmentations make their visual interpretation challenging, especially in relation to observations of “capsid fibers extending from the phage and enveloping the flagellum.” Could the quality of the visualizations be improved by noise removal or manual cleanup? Alternatively, by performing cross-validation on your set of manual annotations, could you employ common evaluation metrics for cryo-ET segmentation such as the DICE and Surface-DICE scores discussed in Lamm et al. [1], in order to assess the accuracy of your trained model?

[1] Lamm et al. MemBrain v2: an end-to-end tool for the analysis of membranes in cryo-electron tomography. biorXiv 2024.

Response: We appreciate the reviewer’s feedback regarding the quality of the automated segmentation results. The observed fragmentation in Figures 5 and S4 (now

Figure 7 and S8) primarily arises from the inherent challenges of segmenting thin, low-contrast structures such as capsid fibers in cryo-electron tomograms. While manual segmentation (as shown in Figure 6) benefits from expert-driven experience, our automated approach is constrained by the variability in fiber visibility and tomographic noise. We have used IsoNet to enhance the clarity of the reconstructed tomograms before segmentation. Although of lesser quality compared to manual segmentation, the obtained results were still good enough to determine the average length of the fibers and their interaction with the flagella in a much less time-consuming manner.

2. An interesting hypothesis is presented that “an initial phage seeks out and attaches to a flagellum, and subsequently, another phage can locate and connect to the first phage via the capsid fibers, thus facilitating its proximity to the bacterial cell and subsequent completion of infection”. Since you have comprehensively segmented tomograms, you might consider highlighting these connections in a figure to provide further evidence.

We appreciate the reviewer’s suggestion. The hypothesis that an initial phage attaches to a flagellum and subsequently facilitates the attachment of additional phages via capsid fibers is indeed a fascinating aspect of our findings. While our segmentation approach allows for detailed tomographic reconstruction, directly visualizing these sequential interactions remains challenging due to resolution constraints and the dynamic nature of phage movement.

Nevertheless, we recognize the value of highlighting these connections to provide additional supporting evidence. To address this, we will include zoomed-in figures to better illustrate the connections between capsid fibers (SI Fig 6).

3. Is there a reason that the phage tails are omitted from the training of the automated segmentation model? The annotation of tails in Figure 4 is helpful in visualizing phage

orientation, and could be similarly helpful in Figure 5.

Response: Thank you for your suggestion. We agree that the phage tail is helpful in visualizing phage orientation. In Figure 6 (previously Fig4), the segmentation was performed manually; however, our neural network was trained specifically to segment phage fibers. Segmenting the tail is less effective because, unlike the fibers, the tail appears as a cylindrical structure with a larger size and greater diameter, making it challenging for this model to distinguish reliably.

The primary purpose of Figure 7 (previously Fig5) is to illustrate the diversity of capsid fibers and provide evidence that phage 771 can use its capsid to "climb" along flagella. Given this focus, we believe that segmenting the phage tail is not essential for the analysis presented in this figure.

Responses to the reviewers' comments:

REVIEWERS' COMMENTS:

Reviewer #1 (Remarks to the Author):

The authors have addressed all my comments.
Congratulations to another solid bacteriophage paper.

Reply: Thank you.

Reviewer #2 (Remarks to the Author):

The authors have addressed all my concerns in the revised manuscript, and it is now ready for publication.

Reply: Thank you.

Reviewer #3 (Remarks to the Author):

The authors have either met or exceeded my expectation in their revisions and I applaud their addition of the neck and tail-tip machinery reconstructions to the manuscript! I endorse the manuscript for publication.

Reply: Thank you.

Reviewer #5 (Remarks to the Author):

I appreciate the authors' clarification of the challenges in segmentation, their use of IsoNet to enhance tomogram interpretability, and the inclusion of a new SI figure highlighting the connections between phages. I believe the machine learning analysis supports their novel structural insights and have no further concerns.

Reply: Thank you.